# STRUCTURE AND BEHAVIOR IN WEIGHT SPACE REPRESENTATION LEARNING

## ABSTRACT

The weights of neural networks (NNs) have recently gained prominence as a new data modality in machine learning, with applications ranging from accuracy and hyperparameter prediction to representation learning or weight generation. One approach to leverage NN weights involves training autoencoders (AEs) with contrastive and reconstruction losses. Indeed, such models can be applied to a wide variety of downstream tasks, and they demonstrate strong predictive performance and low reconstruction error. However, despite the low reconstruction error, these AEs reconconstruct NN models that fail to match the performance of the original ones. In this paper, we identify a limitation of weight-space AEs, specifically highlighting that *structural* weight reconstruction alone fails to capture some features critical for reconstructing high-performing models. To address this issue, we propose a *behavioral* loss for training AEs in weight space. This behavioral loss focuses on the features essential for reconstructing performant models, which are not adequately captured by structural reconstruction. We evaluate the capabilities of AE trained using this novel loss on three different model zoos: we demonstrate that when combining structural and behavioral losses, we can reconstruct and generate models that match the performance of the original models. With our exploration of representation learning in deep weight spaces, we show that a strong synergy exists between structural and behavioral features, and that combining them results in increased performance across all evaluated downstream tasks.

## 1 INTRODUCTION

The weights of trained neural network (NN) models have recently become themselves a domain for research and machine learning, named *weight space learning* (Kofinas et al., 2023; Lim et al., 2023; Navon et al., 2023; Schürholt et al., 2024; Zhou et al., 2023a). Since the weights are structured by the training process, they contain rich information on the data, their generating factors, and also model performance (Unterthiner et al., 2020; Martin et al., 2021). This opens up the opportunity to analyze NNs just by investigating their weights. Understanding the inherent structure in trained weights might further lead to better initializations (Narkhede et al., 2022), model merging (Chou et al., 2018), or identification of lottery tickets (Frankle & Carbin, 2018). However, weight space learning presents several challenges: (i) weight spaces grow with model size and become increasingly sparse; (ii) the weight spaces of different model architectures do not match; (iii) NN architectures contain mutliple invariances and equivariances which result in weight space symmetries.

Several methods have been proposed to address these challenges, for a variety of downstream tasks. Weight derived features have been used for *discriminative* downstream tasks, such as predicting model performance of training hyperparameters (Unterthiner et al., 2020; Eilertsen et al., 2020; Martin et al., 2021; Schürholt et al., 2021). In reverse, other works have developed methods for *generative* downstream tasks, i.e., the generation of synthetic NN weights. For example, HyperNetworks are weight generator models, that use the learning signal from the target model to update the weight-generator (Ha et al., 2017). HyperNetworks have been successfully applied on multiple domains, model sizes, for regular training, architecture search, as well as meta-learning (Zhang et al., 2019; Knyazev et al., 2021; Deutsch, 2018; Ratzlaff & Fuxin, 2019; Zhmoginov et al., 2022).

Recently, an approach has been proposed that can be used for both of these downstream tasks families: extracting information from weights by learning latent representations (dubbed *hyper-representations*) with autoencoders (AEs) (Schürholt et al., 2021). Indeed, the encoder outputs latent representations of NN weights that can be used for discriminative downstream tasks, or fed to the decoder to reconstruct the model. Alternatively, the decoder can be fed synthetic latent representations to generate NN weights (Schürholt et al., 2022a). While these *hyper-representation* AEs show high predictive performance and low reconstruction mean-squared-error (MSE), they appear to lack the fidelity to reconstruct or generate high performance models without further fine-tuning. This is unsurprising since a large component of the composite loss is a MSE reconstruction. MSE reconstruction uses the capacity of the model to predict the mean and as much variation as possible, but may lead to blurry reconstructions and lack of high fidelity (Vincent et al., 2010). Similar observations in computer vision have been addressed by additional image perception losses which force the models to focus on high fidelity features, too (Dosovitskiy & Brox, 2016; Esser et al., 2021).

In this work, we propose a analogous approach for weight space learning to improve reconstruction and generation. We begin by analyzing the error modes of weight-structure reconstruction and identify that *structural* reconstruction misses some of the features essential to reconstructing behaviorally similar models, thus breaking the reconstructed models. Taking inspiration from the image perception loss, we introduce a *behavioral* loss function, and use it for hyper-representation AEs in conjunction with the existing structural loss: we require models to not only have similar weight-structure, but also to behave the same way. We systematically evaluate our approach on discriminative, reconstructive and generative downstream tasks. We find that the addition of a behavioral element to the loss has a minor but positive effect on model analysis tasks. More importantly, we are able to demonstrate that the combination of the structural and behavioral losses dramatically improves the fidelity of the reconstructed models: the models and their respective reconstructions are structurally similar, and they also perform similarly. Finally, we show that this ability to output well-behaved models opens up opportunities to generate synthetic model weights that perform nearly as well as the original ones. In all experiments, we show that there is strong synergy between structural and behavioral aspects of the loss, further confirming that combining those guides the AE to learn representations that are beneficial to model analysis, reconstruction and generation.

## 2    Structural Reconstruction Is Not Enough

**Reconstruction in Weight Space Learning Prioritizes Coarse Features**    Previous work on representation learning of NN weights with AEs has largely focused on reconstructing weights in the structural sense, using the mean square error (MSE) (Schürholt et al., 2021). These representations are effective for discriminative downstream tasks like model performance prediction. However, they struggle in reconstructing functional models without the need for further fine-tuning, even when minimizing the structural reconstruction error (Schürholt et al., 2022a; 2024; Soro et al., 2024). This difficulty may stem from the inductive biases inherent in undercomplete AEs, used in these weight representation learning approaches. Trained with the MSE loss, these AEs often exhibit a bias toward learning coarse, smoothed representations, as MSE minimizes reconstruction error by averaging over fine details (Vincent et al., 2010). Tasks that rely on high-resolution features, like texture recognition, further highlight the limitations of such biases (Zeiler & Fergus, 2014). This smoothing bias may explain the challenges in reconstructing functional models from weight representations.

**Not All Eigenvalues Are Equally Important**    Recent work by Balestriero & LeCun (2024) investigates the usefulness of features learned by AE through reconstruction for image perception tasks. They find that low-eigenvalue features are considerably more useful than high-eigenvalue features. Inspired by this finding, we explore whether a similar phenomenon exists in weight space that might explain the challenges in generating functional models from smooth weight representations. To that end, we train three different model zoos of small convolutional neural networks (CNN) on the SVHN (Netzer et al., 2011), CIFAR-10 (Krizhevsky, 2009) and EuroSAT (Helber et al., 2019) datasets. Following a similar approach to that of Balestriero & LeCun (2024), we represent the weights as a flattened vector. We then perform a projection akin to PCA by selecting certain eigenvectors of the data covariance matrix. We reconstruct the models using the weights after projection and inverse projection, and test the resulting models on their respective test sets.

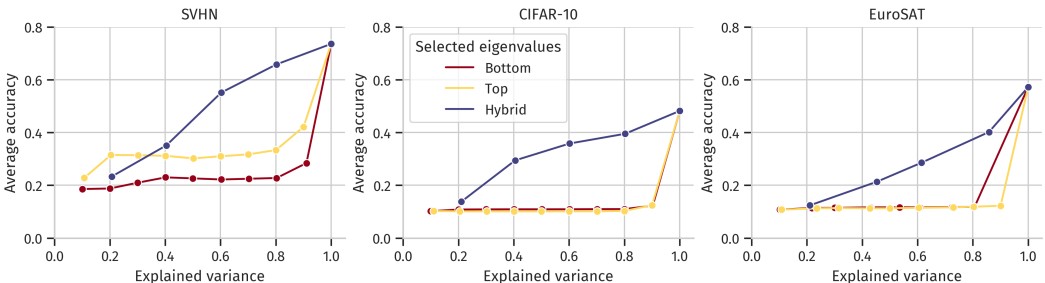

Figure 1: Reconstruction accuracy of convolutional neural networks (CNNs) on SVHN, CIFAR-10, and EuroSAT datasets after projecting and reconstructing the model weights using different sets of eigenvectors. The "Top" and "Bottom" cases retain eigenvectors corresponding to the highest and lowest eigenvalues, respectively. The "Hybrid" case combines the top and bottom eigenvectors, excluding the middle ones. The "Hybrid" approach shows significantly better reconstruction accuracy, even with lower explained variance, showing a strong synergy in combining both top and bottom eigenvectors.

We show the results of our experiment in Figure 1. We consider three different cases: the first two are similar to Balestriero & LeCun (2024) where we keep either the top or bottom eigenvectors that explain a certain amount of variance, and are named "Top" and "Bottom" respectively. In addition, we also include a case where for a target amount of explained variance, we take the top eigenvectors which explain half of it, and the bottom eigenvectors that explain half of it, therefore leaving out the eigenvectors in the middle — we name this approach "Hybrid".

**Weight Reconstruction Requires Low and High Eigenvalues**    On all three datasets, the models reconstructed from either the "Top" or "Bottom" eigenvalues fail to reconstruct accurate models when not all eigenvectors are kept. Conversely, it clearly appears that when using the "Hybrid" eigenvectors, we manage to reconstruct models with a good accuracy, even with relatively low amounts of explained variance. For example, with the SVHN model zoo, models reconstructed with "Hybrid" eigenvectors explaining $60\%$ of the variance show an average accuracy of $55\%$, compared to $31\%$ with the Top and $22\%$ with the Bottom eigenvectors.

The results show that in order to reconstruct models that performs well, we need the features in the subspaces spanned by both the top and bottom eigenvectors, but not so much those in the middle. Schürholt et al. (2022a; 2024); Soro et al. (2024) only use a MSE reconstruction loss in their work, and we know from Balestriero & LeCun (2024) that because of this, the AE network will mostly focus on the data subspace spanned by the top eigenvectors, as it is the most useful for reconstruction. Knowing this, we need further guidance to make the model also focus on the other features essential to reconstruct high-performing NNs. In the following, we show how we develop a behavioral loss to achieve this goal.

## 3    LEARNING FROM STRUCTURE AND BEHAVIOR

Building upon the limitations of autoencoders (AEs) that use a structural loss discussed in Section 2, where the inherent bias towards coarse features leads to reconstructed neural networks (NNs) with degraded functional performance, we propose a *behavioral loss* to enhance the fidelity of the reconstructed models. Inspired by perceptual losses in computer vision (Esser et al., 2021), our behavioral loss emphasizes the functional similarity between the original and reconstructed NNs.

**Composite Representation Learning Loss**    We integrate the behavioral loss into the composite loss originally developed by Schürholt et al. (2021):

$$\mathcal{L} = \gamma \mathcal{L}_C + (1 - \gamma)\left(\beta \mathcal{L}_S + (1 - \beta)\mathcal{L}_B\right), \tag{1}$$

where $\mathcal{L}_C$ is the contrastive loss, promoting discriminative latent representations; $\mathcal{L}_S$ is the structural loss, measuring the parameter-wise difference between the original and reconstructed models; $\mathcal{L}_B$ is the behavioral loss, focusing on the functional discrepancy between the models; and $\gamma, \beta \in [0, 1]$ are hyperparameters balancing the contributions of each component. Since $\mathcal{L}_C$ and $\mathcal{L}_S$ are unchanged compared to previous work, our primary focus is on the behavioral loss $\mathcal{L}_B$.

**Behavioral Loss $\mathcal{L}_B$**    Let: $\theta_j \in \Theta \subset \mathbb{R}^p$ be the parameters of the $j$-th NN from the model zoo, for $j = 1, \ldots, k$; $\hat{\theta}_j = g_w(\theta_j)$ be the reconstructed parameters via the AE with learnable parameters $w$; $f_\theta : \mathcal{X} \to \mathcal{Y}$ denotes an NN that maps on $\mathcal{Y} \subset \mathbb{R}^k$ with weights $\theta$ ($f_{\theta_j}$ and $f_{\hat{\theta}_j}$ are the original and reconstructed NN with parameters $\theta_j$ and $\hat{\theta}_j$ respectively), and $\{x_i\}_{i=1}^n$ be a set of *queries* of input samples from the sample space $\mathcal{X}$. The empirical behavioral loss is defined as:

$$\hat{\mathcal{L}}_b = \frac{1}{2kn} \sum_{j=1}^k \sum_{i=1}^n \left\| f_{\hat{\theta}_j}(x_i) - f_{\theta_j}(x_i) \right\|^2 . \tag{2}$$

This loss measures the discrepancy between the outputs of the original and reconstructed models over the queries, emphasizing functional equivalence.

**Gradient Analysis**    To understand how the behavioral loss influences the AE training differently from the structural loss, we analyze their gradients with respect to the AE parameters $w$. The structural loss is realized as a mean squared error (MSE) over the parameters $\theta$:

$$\hat{\mathcal{L}}_s = \frac{1}{2k} \sum_{j=1}^k \left\| \hat{\theta}_j - \theta_j \right\|^2 . \tag{3}$$

Its gradient with respect to the AE's parameters $w$ is straightforward:

$$\frac{\partial \hat{\mathcal{L}}_s}{\partial w} = \frac{1}{k} \sum_{j=1}^k \Delta\theta_j^\top \frac{\partial \hat{\theta}_j}{\partial w}, \tag{4}$$

where $\Delta\theta_j = \hat{\theta}_j - \theta_j$. The gradient of the behavioral loss with respect to the AE parameters $w$ is:

$$\frac{\partial \hat{\mathcal{L}}_b}{\partial w} = \frac{1}{kn} \sum_{j=1}^k \sum_{i=1}^n \left( f_{\hat{\theta}_j}(x_i) - f_{\theta_j}(x_i) \right)^\top \frac{\partial f_{\hat{\theta}_j}(x_i)}{\partial \hat{\theta}_j} \frac{\partial \hat{\theta}_j}{\partial w}, \tag{5}$$

where we abuse the partial derivative notation $\frac{\partial f_{\hat{\theta}_j}(x_i)}{\partial \hat{\theta}_j}$ to mean taking the partial derivative with respect to $\theta$ evaluated at $\hat{\theta}_j$. Assuming that the reconstructed parameters are close to the original ones (i.e., $\Delta\theta_j$ is small), which is a justified assumption when $||\hat{\theta}_j - \theta_j||_2^2$ is part of the loss function and the AE is sufficiently expressive, we can approximate $f_{\hat{\theta}_j}(x_i)$ using a first-order Taylor expansion around $\theta_j$:

$$f_{\hat{\theta}_j}(x_i) \approx f_{\theta_j}(x_i) + J_{\theta_j}(x_i)\Delta\theta_j, \tag{6}$$

where $J_{\theta_j}(x_i) = \frac{\partial f_{\theta_j}(x_i)}{\partial \theta_j}$, with $J_{\theta_j}(x_i)^\top \in \mathbb{R}^{p \times k}$, is the Jacobian of the model output with respect to its parameters. Substituting into the gradient:

$$\frac{\partial \hat{\mathcal{L}}_b}{\partial w} \approx \frac{1}{kn} \sum_{j=1}^k \sum_{i=1}^n \left( J_{\theta_j}(x_i)\Delta\theta_j \right)^\top J_{\hat{\theta}_j}(x_i) \frac{\partial \hat{\theta}_j}{\partial w}$$

$$= \frac{1}{k} \sum_{j=1}^k \Delta\theta_j^\top \left( \frac{1}{n} \sum_{i=1}^n J_{\theta_j}(x_i)^\top J_{\hat{\theta}_j}(x_i) \right) \frac{\partial \hat{\theta}_j}{\partial w}. \tag{7}$$

This allows us to approximate the gradient of the behavioral loss as:

$$\frac{\partial \hat{\mathcal{L}}_b}{\partial w} \approx \frac{1}{k} \sum_{j=1}^k \Delta\theta_j^\top F_j \frac{\partial \hat{\theta}_j}{\partial w}, \tag{8}$$

where $F_j = \frac{1}{n} \sum_{i=1}^n J_{\theta_j}(x_i)^\top J_{\hat{\theta}_j}(x_i)$.

**Behavioral Gradients Modulate Structural Gradients with Weight Importance**  Comparing the behavioral loss gradient (Eq. 8) to the structural loss gradient (Eq. 4) shows that both depend on $\Delta\theta_j$, the difference between the original and reconstructed parameters, and $\frac{\partial\hat{\theta}_j}{\partial w}$. However, while the structural loss gradient measures the linear alignment between $\Delta\theta_j$ and $\frac{\partial\hat{\theta}_j}{\partial w}$, the behavioral loss gradient measures the linear alignment between $\Delta\theta_j$ and a vector $F_j\frac{\partial\hat{\theta}_j}{\partial w}$. $F_j$ plays a key role in the gradient updates because it provides information on the average sensitivity of the original and reconstructed NNs to changes in the weights, as well as the average linear alignment between their corresponding gradients w.r.t. the parameters.

**The Choice of Queries Impacts the Learned Representations**  Two NNs trained on the same dataset with similar performance on the domain of the training data can have different behavior outside that domain. Similarly, since the behavioral loss depends on the queries used to compute it, the choice of those will influence which aspects of the behavior of the original models will be reconstructed. If the queries used for the behavioral loss come from a different domain than the one of the training data, the AE will attempt to match the behavior of the original and reconstructed model on parts of the domain where the NN from the zoo did not have training data, and where its behavior is therefore ill-defined. This implies that the reconstructed and original model will match performance on parts of the domain that is of no interest, while on the parts of interest the performance might be arbitrarily different. Hence, the choice of the queries plays an important role in the performance of the reconstructed models. In Appendix D.3 we explore the hypothesis that randomly generated queries lead to poor performing reconstructed models, while queries from the same or similar distribution as the training data lead to good performance of the reconstructed models.

## 4 EXPERIMENTS

In the following, we build on the conclusions of Section 3 and experimentally evaluate whether the inclusion of a behavioral element to the loss function leads to better performance on a selection of downstream tasks from the literature. We first describe our general experimental setup, and then compare different variants of our loss functions for each individual downstream task: discriminative, reconstructive, and generative.

### 4.1 EXPERIMENTAL SETUP

**Model Zoos**  In this work, we understand a model zoo as a structured population of models using the same architecture and trained on the same data. We train three different model zoos of convolutional neural networks (CNN), on the SVHN (Netzer et al., 2011), CIFAR-10 (Krizhevsky, 2009) and EuroSAT (Helber et al., 2019) datasets. Every model zoo is built with the same grid of hyperparameters. In total, every zoo is composed of $1,200$ models, with $10,853$ parameters each, trained over 50 epochs. We provide additional details about the model zoos generation, their architecture and their hyperparameters in Appendix A. The model zoos are randomly separated into disjoint train, validation and test splits with respective proportions $\{80\%, 5\%, 15\%\}$.

**Loss Functions**  We use the composite loss function defined in Equation 1. It combines three elements: a contrastive loss in latent space $\mathcal{L}_C$, and two losses in reconstructed space, one structural $\mathcal{L}_S$, one behavioral $\mathcal{L}_B$. Their relative weights are controlled by two hyperparameters $\gamma, \beta \in [0, 1]$. When $\gamma = 0$ the contrastive loss is not used; when $\beta = 0$ the structural loss is not used; and when $\beta = 1$ the behavioral loss is not used. When using both the contrastive loss and a reconstructive loss, we set $\gamma = 0.05$, as in the original SANE implementation. When using both the structural and behavioral loss, we set $\beta = 0.1$. This value follows from an exploration of possible values on our validation set and has shown to perform best, as detailed in Appendix D.6. The case where $\gamma = 0.05$ and $\beta = 1$ corresponds to the existing SANE implementation and serves as our baseline. We implement the contrastive loss $\mathcal{L}_C$ the same way as in SANE, with NTXent (Sohn, 2016) where the augmentations used are permutations of weights that do not alter the behavior of the NN. $\mathcal{L}_S$ and $\mathcal{L}_B$ are implemented using the mean-squared-error loss, which is equivalent to the $L^2$ loss up to a constant factor and a square root.

Table 1: Discriminative downstream tasks performance. We predict the test accuracy and generalization gap of our models based on their latent representation, using a linear probe. We give the $R^2$ score for predictions on the held-out test split. We express the 'Improvement' as the $R^2$ score of $\mathcal{L}_C + \mathcal{L}_S + \mathcal{L}_B$ minus the score of the baseline $\mathcal{L}_C + \mathcal{L}_S$. In all cases, the most performant loss combination includes both the structural loss $\mathcal{L}_S$ and the behavioral loss $\mathcal{L}_B$.

| LOSSES | TEST ACCURACY | | | GENERALIZATION GAP | | |
|---|---|---|---|---|---|---|
| | SVHN | CIFAR-10 | EUROSAT | SVHN | CIFAR-10 | EUROSAT |
| $\mathcal{L}_C + \mathcal{L}_S$ (BASELINE) | 0.742 | 0.890 | 0.957 | 0.347 | 0.700 | 0.465 |
| $\mathcal{L}_B$ | 0.538 | 0.771 | 0.901 | 0.286 | 0.576 | 0.296 |
| $\mathcal{L}_C + \mathcal{L}_B$ | 0.752 | 0.893 | 0.950 | 0.337 | 0.710 | 0.482 |
| $\mathcal{L}_S + \mathcal{L}_B$ | **0.887** | 0.939 | 0.966 | **0.378** | **0.785** | 0.484 |
| $\mathcal{L}_C + \mathcal{L}_S + \mathcal{L}_B$ | 0.886 | **0.947** | **0.969** | 0.368 | **0.785** | **0.529** |
| IMPROVEMENT | 0.144 | 0.056 | 0.012 | 0.021 | 0.085 | 0.063 |

**Queries for $\mathcal{L}_B$**  Computing the behavioral loss $\mathcal{L}_B$ requires us to use some queries to feed to both the original and reconstructed models, so as to compare their outputs. For every batch, we sample $n_{queries} = 256$ images from the training set used to train the corresponding model zoo.

**Hyper-Representation AEs**  We use SANE, the implementation of hyper-representation AEs by Schürholt et al. (2024), where the weights of the original model are tokenized, then fed to an encoder that generates one latent representation per token. A projection head is used for the contrastive loss $\mathcal{L}_C$, whereas all embeddings are fed to the decoder. With an original token length of $289$ and an embedded dimension of $64$, our compression ratio is $4.52$. While SANE allows the use of windows of tokens rather than entire models, we systematically feed and reconstruct an entire model at once; we discuss this limitation in more details in Section 6. We train our hyper-representation models on the train split of the corresponding model zoo, using the checkpoints corresponding to training epochs $\{20, 30, 40, 50\}$. We provide additional details about hyper-representation AEs training in Appendix B. Additionally, we extend these results with some ablation experiments in Appendix D.

## 4.2 COMBINING STRUCTURE AND BEHAVIOR INCREASES PERFORMANCE FOR EVERY TASK

In the following, we evaluate the performance of the hyper-representation AEs trained with different values of $\gamma$ and $\beta$ for three different kinds of downstream tasks: discriminative, reconstructive and generative. Throughout, to facilitate readability, we refer to the presence or absence of the individual elements of the composite loss as described in Section 4.1, rather than the specific values of those hyperparameters. For each downstream task, we first describe the task and how it is evaluated, and then present the empirical results.

### 4.2.1 DISCRIMINATIVE DOWNSTREAM TASKS

We consider two different discriminative downstream tasks, where we try to predict either a model's test accuracy or its generalization gap (defined as the difference between the train and test accuracies). To do so, we compute its hyper-representation using AEs trained with different elements of the composite loss. Following existing evaluation setups from the literature, we average all embedded tokens together into a $64$ dimensional "center of gravity" of the embeddings, and use a linear probe on top of it. The probe is trained on the models in the train split, and evaluated on the held-out test split. We show results in Table 1.

The results show that although we built mostly to improve reconstructive accuracy, our behavioral loss also improves the performance of our discriminative downstream tasks: in all cases, the most performant loss combination includes both $\mathcal{L}_S$ and $\mathcal{L}_B$. Since we use linear probes, that means that the behavioral loss plays a part in better structuring the latent space, and that combining it with the structural loss synergizes well.

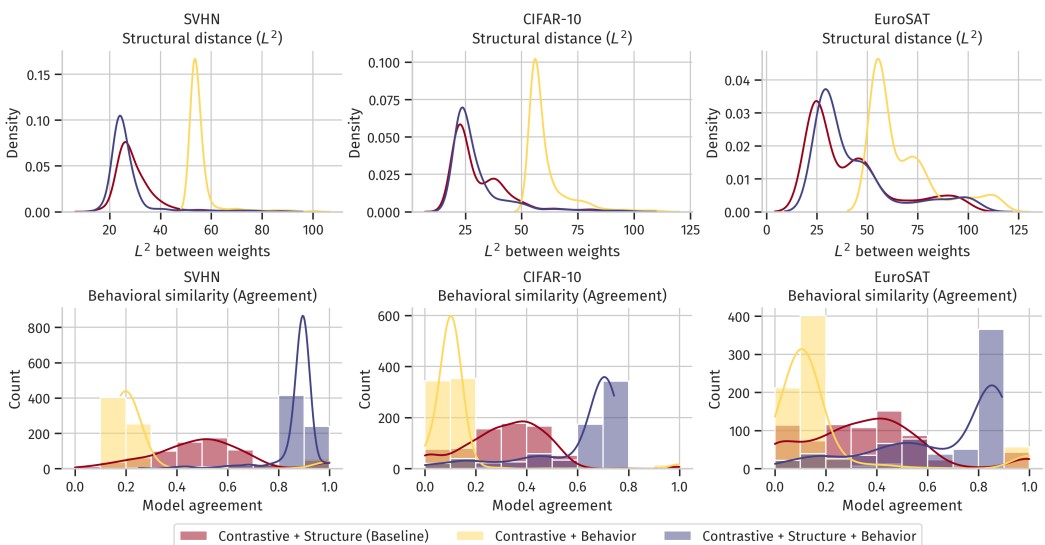

Figure 2: Evaluation of the reconstructive downstream tasks, pairwise between models from the model zoo and their reconstructions, depending on the losses used to train the hyper-representation AE. Each column represents one of our model zoos. On the top row, we show the distribution of pairwise structural $L^2$ distances. On the bottom row, we show the distribution of the pairwise behavioral similarities, measured with model agreement. On the structural side, we see that using the structural loss is sufficient to concentrate most pairwise distances around some low value. On the behavioral side, we see that using the behavioral loss only yields the worst performance, even compared to the structural loss. Using both the structural and behavioral losses is necessary to achieve high levels of agreement. Most models show high levels of agreements, but since a few show low levels of agreement the standard deviation shown in Table 5 can be high.

### 4.2.2 Reconstructive Downstream Tasks

With the addition of the behavioral loss, we target reconstructed models to be similar to the originals, both in structure and behavior. Our model zoos are diverse, and contain both high- and low-performing models: we expect a poorly performing model to be reconstructed as a poorly performing model, and conversely for well-performing models. We evaluate the fidelity of reconstructed models first pairwise, then in distribution over the whole test split. All metrics are computed over the test split of the model zoos, which only includes models not seen by the hyper-representation AE at training time. We show results for pairwise evaluation in Figure 2, and report more detailed results in Appendix Table 5.

We first assess the structural reconstruction fidelity by comparing the pairwise $L^2$ distances between a model weights and its reconstruction. We note that using $\mathcal{L}_S$ is sufficient and necessary to get low structural reconstruction errors, with large differences between models that use it and those that do not. We then measure the behavioral distance between models and their reconstruction using model agreement. Conversely to the structural distance, we note that using the behavioral loss does not guarantee high model agreement: models trained with $\mathcal{L}_B$ but without $\mathcal{L}_S$ show both the highest structural error and lowest model agreement. At the same time, models trained with both $\mathcal{L}_S$ and $\mathcal{L}_B$ show much higher agreement than all others. In particular, when comparing those models to our baseline that uses $\mathcal{L}_C$ and $\mathcal{L}_S$, we see that including $\mathcal{L}_B$ is essential to achieve a reconstruction that is behaviorally similar to the original model.

We compare the distributions of model accuracies for models from the zoos and their reconstructions in Figure 3. We compare the baseline AE that uses $\mathcal{L}_C$ and $\mathcal{L}_S$ only to our model that is also trained with $\mathcal{L}_B$. In all cases, the baseline fails to reconstruct models of performance comparable to the ones in the original zoo. On the other hand, when including $\mathcal{L}_B$, reconstructed models are very performant — at the same time, there seems to be a bias towards reconstructing models with higher performance than the original when the latter is not among the very best models.

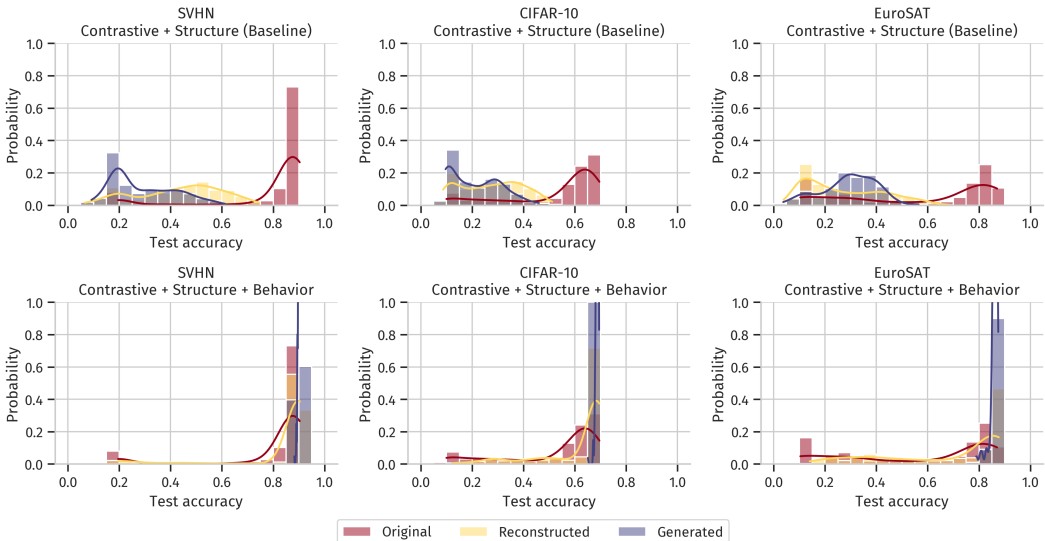

Figure 3: Evaluation of the reconstructive and generative downstream tasks, shown as distributions of the test accuracy of different models, depending on whether they are part of the original model zoo, reconstructions of models from that model zoo, or generated models. Each column represents one of our model zoos, while the row show what loss has been used to train the specific hyper-representation model. The top row shows results for our baseline, that uses the contrastive $\mathcal{L}_C$ and structural $\mathcal{L}_S$ losses. The bottom row represents our hyper-representation AEs, which in addition are also trained with a behavioral loss $\mathcal{L}_S$. We note that for the baseline, neither the reconstructed nor the generated models can match the performance of the original models. On the other hand, when adding a behavioral element to the loss, they match the performance of the most accurate models from the original zoo.

### 4.2.3 GENERATIVE DOWNSTREAM TASKS

In addition to reconstructing exisiting models, fixing the hyper-representation AE's ability to output performant model weights opens up the way for generating new model weights. In their existing work, Schürholt et al. (2022a) generate models using their hyper-representations, but their performance is limited and they have to be re-trained for a few epochs to achieve performance comparable to models in the original zoo. In the following, we evaluate whether hyper-representation AEs trained with a behavioral loss can perform better at model weights generation.

To generate model weights, we first need to generate corresponding synthetic latent representations. To do so, we select anchor models from the model zoo that are themselves well-performing (more details in Appendix B) and compute their latent representations. With the goal of mitigating the curse of dimensionality, we then use PCA to project those in a lower dimensional space of size 32, and use a kernel density estimate (KDE) to model the distribution of the projected embeddings. For the KDE, we further assume that the coordinates are orthogonal to each other. We generate new data points by sampling from that KDE, and inverse project back into the latent representation space. Finally, we feed these synthetic hyper-representations to the decoder and evaluate the resulting models. We show results in Figure 3.

We first note that when using the baseline hyper-representation AE, the generated models tend to be as good as or slightly worse than the reconstructed models. This level of performance is low compared to the anchors we use for generation, which for all three datasets have more than 60% test accuracy. On the other hand, when adding our behavioral element to the loss function, we see that the generated models have a very high test accuracy, very close to the best models in the model zoo, as reported in Table 2. Their performance seem to match that of the best reconstructed models. Finally, we show in Appendix Table 6 that generated models are somewhat diverse, although less than the original anchors. This confirms that we do not generate identical models.

Table 2: Maximum performance of selected models. 'Zoo' describes models from the original model zoo; 'Recon.' includes the reconstructions of the models from the test split of the model zoo; 'Gener.' represents the models synthetically generated as described in Section 4.2. '$\Delta_{Acc}$ Rec.' and '$\Delta_{Acc}$ Gen.' refer to the difference in performance between the best models in respectively the reconstructed and generated models on one side, and the original models on the other side. For the baseline that only uses the contrastive $\mathcal{L}_C$ and structural $\mathcal{L}_S$ losses, reconstruction and generation do not manage to build models of comparable performance with the original zoo. Conversely, when adding the behavioral loss $\mathcal{L}_B$, we achieve a maximum performance very close to that of the original zoo.

| | LOSSES | ZOO | RECON. | $\Delta_{Acc}$ REC. | GENER. | $\Delta_{Acc}$ GEN. |
|---|---|---|---|---|---|---|
| SVHN | $\mathcal{L}_C + \mathcal{L}_S$ (BASELINE) | 91.0% | 74.5% | -16.5% | 61.3% | -29.7% |
| | $\mathcal{L}_C + \mathcal{L}_S + \mathcal{L}_B$ | 91.0% | 90.4% | **-0.6%** | 90.4% | **-0.6%** |
| CIFAR-10 | $\mathcal{L}_C + \mathcal{L}_S$ (BASELINE) | 70.1% | 51.2% | -18.9% | 46.0% | -24.1% |
| | $\mathcal{L}_C + \mathcal{L}_S + \mathcal{L}_B$ | 70.1% | 69.5% | **-0.6%** | 69.5% | **-0.6%** |
| EUROSAT | $\mathcal{L}_C + \mathcal{L}_S$ (BASELINE) | 88.5% | 68.6% | -19.9% | 56.5% | -32.0% |
| | $\mathcal{L}_C + \mathcal{L}_S + \mathcal{L}_B$ | 88.5% | 87.7% | **-0.8%** | 87.5% | **-1.0%** |

## 5 RELATED WORK

**Weight Space Representation Learning**   Recent work in representation learning on NN weights has led to various approaches for analyzing and generating models weights. Hyper-Representations (Schürholt et al., 2021; 2022b;a; Soro et al., 2024) use an encoder-decoder architecture with contrastive guidance to learn weight representations for property prediction and model generation. Using a different learning task for weight generation, other methods employ diffusion on weights (Peebles et al., 2022; Wang et al., 2024; Jin et al., 2024). Our work differs by focusing on understanding and mitigating the inductive biases in weight space learning, particularly for AE-based approaches. Similar to our behavioral loss, HyperNetworks use the learning feedback from the target models to generate their weights (Ha et al., 2017; Knyazev et al., 2021; 2023). Graph representation methods (Zhang et al., 2019; Kofinas et al., 2023; Lim et al., 2023), Neural Functionals (Zhou et al., 2023a;b; 2024) and related approaches like Deep Weight Space (DWS)(Navon et al., 2023; Zhang et al., 2023) learn equivariant or invariant representations of weights. While these methods incorporate geometric priors of the weight space in encoder or decoder models, we focus in this work on AE-based approaches, as they cover the largest breadth of downstream tasks. Augmentations specific to weight space learning have also been developed (Shamsian et al., 2024). As mentioned in Section 4.1, our experiments focus on augmentations that do not alter the behavior of the NN, and therefore do not include such augmentations.

**Probing-Based Losses in Weight Space**   Other works have leveraged losses that are based on the response of models given a set of queries, first of which are HyperNetworks (Ha et al., 2017). De Luigi et al. (2023) use weight-space auto-encoders, but they feed the queries directly to the decoder which is then trained on the true labels. Navon et al. (2023) propose an architecture that can be used in many different setups, one of them being domain adaptation, where they use the loss on the new domain. More closely related to our work, Herrmann et al. (2024) leverage behavioral losses (which they name functionalist) to analyze recurrent neural networks (RNNs) with high performance. Their work differs from ours as they use the decoder as an emulator of the function represented by the original model, whereas we directly reconstruct the original model. Navon et al. (2024) use a composite loss in their deep weight space alignment experiment setup, with elements closely related to structure and behavior but tailored to their particular use-case. For example, they use probing-based self-supervised losses as a way to measure linear mode connectivity between two models, rather than to compare their individual behavior. Our work's exploration of the synergy between structure and behavior gives insights in how incorporating both structural and behavioral loss elements can have contributed to the good performance of their model.

**Generation of Neural Networks for a Fixed Architecture**   Different attempts at generating NN weights for a given architecture have been proposed, such as random generation of weights without explicit use of data (Schrauwen et al., 2007; Timotheou, 2010), pruning of overparameterized randomly generated networks (Malach et al., 2020), learning weights as a function of the data via imposing a probability measure on the dataset (Bolager et al., 2024), reconstruction of weights by evaluating the network on a specific dataset (Fornasier et al., 2022), etc. Our approach differs because it does not use labelled data, nor does it generates the weights in a data-agnostic sense.

**Inductive Biases of AEs**   AEs trained with mean squared error (MSE) loss often exhibit an inductive bias toward learning coarse, smoothed representations, which can result in the loss of fine-grained details important for certain tasks (Alain & Bengio, 2014; Vincent et al., 2010). However, tasks that rely on high-resolution features, such as texture recognition and fine-grained classification, require the retention of these fine details for optimal performance (Zeiler & Fergus, 2014; Gatys et al., 2016). To address these limitations, alternative approaches like perceptual losses (Johnson et al., 2016), adversarial training (Makhzani et al., 2016), and variational methods (Kingma & Welling, 2013) have been proposed to preserve fine details in reconstructions. Similar challenges arise in NN weight space learning, where coarse weight reconstructions may lead to suboptimal model performance due to the loss of important structural details in the network's weights (Li et al., 2016). We take inspiration from the existing work in the computer vision domain to build a behavioral loss with the goal of mitigating this issue in weight space.

## 6 DISCUSSION

**Limitations**   This paper highlights the importance of considering both structure and behavior when training AEs in weight space, as both are essential for reconstructing and generating high-performing models. However, since scaling tranformer-based architectures to large sequences is challenging, our exploration is limited to smaller models. Our work focuses on validating our findings and the concept of a behavioral loss: we defer its implementation for larger NNs to future work. Another limitation is the computational overhead, as we need to reconstruct and test generated models at each training step. In Appendix D.4 however, we show that for comparable computing time, using structure and behavior still outperforms fully-structural approaches. Finally, as shown in Appendix D.3, using a proper set of queries that is close to that used to train the model zoo is important. This means that conversely to the fully-structural approach, there is a need for unlabeled data samples when using the behavioral loss.

**Conclusion**   Our work presents an in-depth exploration of the weight-space modality in the context of representation learning, presenting strong synergies in combining structural and behavioral signals when learning from populations of trained NNs. We first show that to reconstruct accurate models, weight-space AEs need to focus on features that are spanned by both the top and bottom eigenvectors of the data covariance matrix, while those in the middle matter less. Since fully structural AEs tend to mostly focus on features spanned by the top eigenvectors, we build a behavioral loss function to guide the learning of weight-space AEs towards the other features that are essential to reconstruct high-performing models, and theoretically explore how it differs from a purely structural loss. Finally, we demonstrate experimentally that adding a behavioral element into the loss function of weight-space AEs synergizes with the structural element, with the resulting models outperforming the purely structural baseline for discriminative, reconstructive as well as generative downstream tasks. More generally, our work shows that training self-supervised models in weight space requires a balance between structural and behavioral features to perform well. Our analysis uncovers hitherto unknown insights and opens up exciting research opportunities in the domain of weight-space representation learning.

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

## A  MODEL ZOOS GENERATION

All the models in our zoos have the same architecture, described in Table 3. For the EuroSAT dataset, we first resize the images to $32 \times 32$ so that for all three datasets, inputs are of dimension $32 \times 32 \times 3$ and outputs of dimension 10. For all datasets, we use standardization with the mean and standard deviations for ImageNet. We do not use augmentations.

Table 3: CNN architecture used for our experiments.

| LAYER | HYPERPARAMETER | VALUE |
|---|---|---|
| Convolutional 1 | Channels in | 3 |
| | Channels out | 16 |
| | Kernel size | 3 |
| MaxPool 1 | Kernel size | 2 |
| | Stride | 2 |
| ReLU 1 | | |
| Convolutional 2 | Channels in | 16 |
| | Channels out | 32 |
| | Kernel size | 3 |
| MaxPool 2 | Kernel size | 2 |
| | Stride | 2 |
| ReLU 2 | | |
| Convolutional 3 | Channels in | 32 |
| | Channels out | 15 |
| | Kernel size | 3 |
| MaxPool 3 | Kernel size | 2 |
| | Stride | 2 |
| ReLU 3 | | |
| Flatten | | |
| Linear 1 | Dimension in | 60 |
| | Dimension out | 20 |
| ReLU 4 | | |
| Linear 2 | Dimension in | 20 |
| | Dimension out | 10 |

Models are trained for 50 epochs, with a batch size of 32. We use the Adam optimizer. To generate diverse models within our zoo, we vary other hyperparameters, as described in Table 4. This results in a total of $1,200$ models per zoo, with one checkpoint for each training epoch. In Figure 4, we show the distribution of model test accuracies in all three zoos, throughout the training process.

Table 4: Training hyperparameters for our model zoos. Kaiming initializations refer to work by He et al. (2015).

| HYPERPARAMETER | VALUES |
|---|---|
| Initialization | Uniform, Normal, Kaiming Uniform, Kaiming Normal |
| Learning rate | $1e-4, 1e-4, 2.5e-4, 5e-4, 7.5e-4, 1e-3$ |
| Weight decay | $1e-4, 5e-4, 1e-3$ |
| Seed | $0, 1, ..., 19$ |

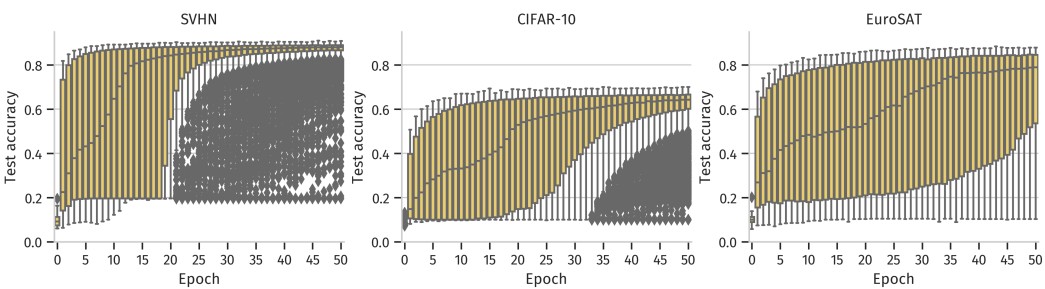

Figure 4: Distribution of the model test accuracies in our three model zoos, by training epoch.

## B  HYPER-REPRESENTATION MODEL TRAINING

**Pre-processing**   While the original `SANE` implementation uses several pre-processing steps such as weight alignment with Git Re-Basin (Ainsworth et al., 2022) or standardization, we find empirically that such steps are not necessary anymore when using $\mathcal{L}_B$. We therefore only use a single pre-processing step, which is the pre-computation of behaviorally equivalent weight permutations of models in our zoo, which are used as augmentations and in the context of the contrastive loss $\mathcal{L}_C$.

**Hyperparameters**   We train our hyper-representation transformer encoder and decoder with the following hyper-parameters: an input dimension of 289, an output dimension of 64, and a model dimension (`d_model`) of 256. The encoder was configured with 8 attention heads (`num_heads`) and comprised 8 layers (`num_encoder_layers`). We use a batch size of 64, and a weight decay of $3e-9$. After grid optimization, we select a learning rate of $1e-4$ for our baseline ($\mathcal{L}_C + \mathcal{L}_S$), and $1e-5$ for all others (i.e., those that use $\mathcal{L}_B$). Training is done for 100 epochs.

**Anchors for the generative downstream tasks**   Regarding the selection of anchor models for the generative downstream task, we select only models with good performance compared to the rest of the zoo. As shown in Figure 4, the distribution of accuracies varies a lot depending on the zoo we take into consideration. For this reason, we choose different test accuracy thresholds for each zoo: 80% for SVHN, 60% for CIFAR-10 and 70% for EuroSAT.

**Computational load**   We ran all experiments on NVIDIA Tesla V100 GPU. We train for a total of 100 epochs, and on a training set composed of $4,080$ checkpoints ($1,020$ different models, training epochs $\{20, 30, 40, 50\}$). When the behavioral loss is not used, training takes around $4,400$ seconds or a little less than 1 hour and 15 minutes, while when using the behavioral loss, training takes around $8,300$ seconds or a little more than 2 hours and 15 minutes. There is a notable difference because instead of only comparing tokens to tokens, the reconstructed tokens have to be converted back into a usable model, and then a forward pass must be done over the reconstructed model. Fortunately, computing this for each model in a batch is parallelized, hence why the difference in computation time is still under a factor of 2. While indeed more expensive than the fully-structural approach, we deem the trade-off for increased performance acceptable.

## C  ADDITIONAL RESULTS

In this Section, we show additional results in Table format that complement those shown in Section 4.2. In Table 5, we show structural distance and behavioral similarity between models and their reconstructions, pairwise. In Table 6, we study the diversity of the generated models, both in terms of structure and behavior.

Table 5: Reconstructive downstream tasks performance. We evaluate structural reconstruction with the average $L^2$ distances between the weights of test split models and their reconstructions. We evaluate behavioral reconstruction with the average classification agreement between test split models and their reconstructions. Standard deviation is indicated between parentheses. BL indicates the baseline. While using the structural loss $\mathcal{L}_S$ is sufficient and necessary to get low strucural reconstruction distance, the behavioral loss $\mathcal{L}_B$ alone does not suffice to reconstruct behaviorally close models. Only the combination of $\mathcal{L}_S$ and $\mathcal{L}_B$ allows the reconstruction of models that are close both in terms of structure and behavior.

| LOSSES | STRUCTURE ($L^2$ DISTANCE) | | | BEHAVIOR (AGREEMENT) | | |
|---|---|---|---|---|---|---|
| | SVHN | CIFAR-10 | EuroSAT | SVHN | CIFAR-10 | EuroSAT |
| $\mathcal{L}_C + \mathcal{L}_S$ (BL) | 30.7 (±10) | 31.3 (±13) | **41.4 (±22)** | 50.0% (±20%) | 31.9% (±16%) | 35.9% (±24%) |
| $\mathcal{L}_B$ | 57.8 (±6) | 59.5 (±9) | 67.5 (±17) | 26.6% (±22%) | 10.3% (±8%) | 19.2% (±25%) |
| $\mathcal{L}_C + \mathcal{L}_B$ | 55.7 (±7) | 60.7 (±9) | 67.4 (±17) | 26.6% (±22%) | 11.8% (±14%) | 18.5% (±25%) |
| $\mathcal{L}_S + \mathcal{L}_B$ | **26.9 (±10)** | **29.6 (±13)** | 45.5 (±21) | 86.1% (±11%) | 59.4% (±20%) | **66.5% (±25%)** |
| $\mathcal{L}_C + \mathcal{L}_S + \mathcal{L}_B$ | 27.1 (±10) | 30.0 (±13) | 43.6 (±21) | **87.0% (±9%)** | **59.6% (±19%)** | 65.8% (±25%) |

Table 6: Diversity of generated models, expressed as the mean pairwise $L^2$ distance between all models in the selected set, computed either on weights (structure) or predictions (behavior). Anchors are the models from the model zoo used as a basis to generate synthetic latent representations. In all cases, there is some diversity in then generated models, showing that we do not generate identical models. However, when using the behavioral loss $\mathcal{L}_B$, we see that both the structural and behavioral diversity of generated models is lower.

| LOSSES | STRUCTURE ($L^2$ DISTANCE) | | | BEHAVIOR ($L^2$ DISTANCE) | | |
|---|---|---|---|---|---|---|
| | SVHN | CIFAR-10 | EuroSAT | SVHN | CIFAR-10 | EuroSAT |
| ANCHORS | 23.3 (±4) | 22.9 (±4) | 23.0 (±5) | 3.6 (±0) | 4.7 (±0) | 4.2 (±1) |
| $\mathcal{L}_C + \mathcal{L}_S$ (BASELINE) | 13.4 (±3) | 14.9 (±4) | 16.8 (±5) | 4.1 (±2) | 4.9 (±2) | 8.7 (±2) |
| $\mathcal{L}_B$ | 1.6 (±0) | 3.5 (±1) | 6.3 (±1) | 0.0 (±0) | 0.2 (±0) | 0.1 (±0) |
| $\mathcal{L}_C + \mathcal{L}_B$ | 1.7 (±0) | 1.5 (±0) | 3.0 (±1) | 0.0 (±0) | 0.1 (±0) | 0.1 (±0) |
| $\mathcal{L}_S + \mathcal{L}_B$ | 1.6 (±0) | 1.5 (±0) | 2.8 (±1) | 0.6 (±0) | 0.8 (±0) | 1.2 (±1) |
| $\mathcal{L}_C + \mathcal{L}_S + \mathcal{L}_B$ | 1.5 (±0) | 1.3 (±0) | 2.1 (±1) | 0.7 (±0) | 0.7 (±0) | 1.1 (±1) |

# D  ABLATION EXPERIMENTS

## D.1  CONTRASTIVE LOSS IS STILL USEFUL

In Section 4.2, we show that when combining the structural $\mathcal{L}_S$ and behavioral $\mathcal{L}_B$ losses, the performance of all downstream tasks is improved. This raises an additional question: is using the contrastive loss $\mathcal{L}_C$ still relevant? The results shown in Tables 1 and 5 look inconclusive in that regard. As $\mathcal{L}_C$ focuses in structuring the latent space and has little influence over the decoder, we mostly evaluate it with regard to the discriminative downstream tasks that take place in the latent space, in Figure 5.

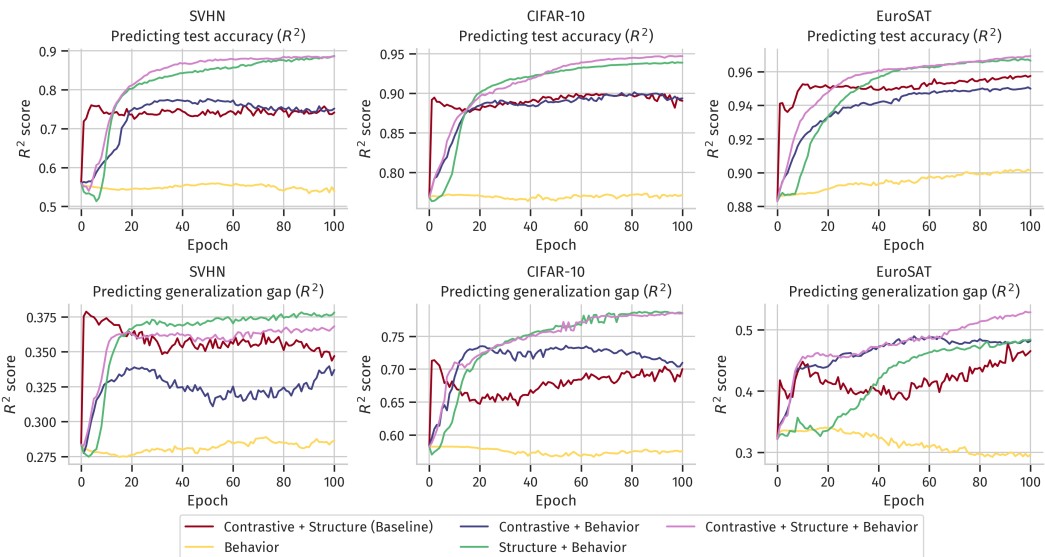

Figure 5: Comparison of the performance of the discriminative downstream tasks by training epoch of SANE. We first note that both models that use $\mathcal{L}_S$ and $\mathcal{L}_B$, show a more stable increase in performance with each epoch, and mostly outperform other models. Then, comparing the model that uses $\mathcal{L}_S + \mathcal{L}_B$ with the one that uses $\mathcal{L}_C + \mathcal{L}_S + \mathcal{L}_B$, we qualitatively note that in most cases, performance grows faster with the number of epochs.

We compare both models that use $\mathcal{L}_S$ and $\mathcal{L}_B$: the one that uses $\mathcal{L}_C$ and the one that does not. Qualitatively, the performance of the model that uses $\mathcal{L}_C$ grows faster with the number of training epochs compared to that of the model that does not use $\mathcal{L}_C$. Additionally, we note from Table 1 that when predicting the generalization gap for the EuroSAT zoo, the model that uses $\mathcal{L}_C$ outperforms the one that does not by $0.045$, whereas the largest difference in performance in the other direction happens for predicting the generalization gap for the SVHN zoo, and is only of $0.01$, or $4.5$ times inferior. Although relatively weak, empirical evidence seems to indicate that using the contrastive loss $\mathcal{L}_C$ together with both structure $\mathcal{L}_S$ and behavior $\mathcal{L}_B$ remains relevant.

## D.2  MSE IS THE MOST STABLE BEHAVIORAL LOSS

In Section 4.1, we use a MSE loss (which is equivalent to the $L^2$ distance up to a constant factor and a square root) over the predictions of a model and that of its reconstruction. There are, however, other losses that could be used in that context. We explore those empirically in this Section. In particular, we tested using either a cross-entropy loss, or a distillation loss (Hinton, 2015) with a temperature of $2$. We explore these on hyper-representations trained with all three elements of the composite loss: contrastive $\mathcal{L}_C$, structural $\mathcal{L}_S$ and behavioral $\mathcal{L}_B$. We show the resulting training losses in Figure 6.

We observe that despite our best attempts, both the cross-entropy and distillation losses are numerically unstable for at least one model zoo, whereas the MSE loss remains stable and decreasing in all cases. This could not be resolved by changing the learning rate. For this reason, and because the

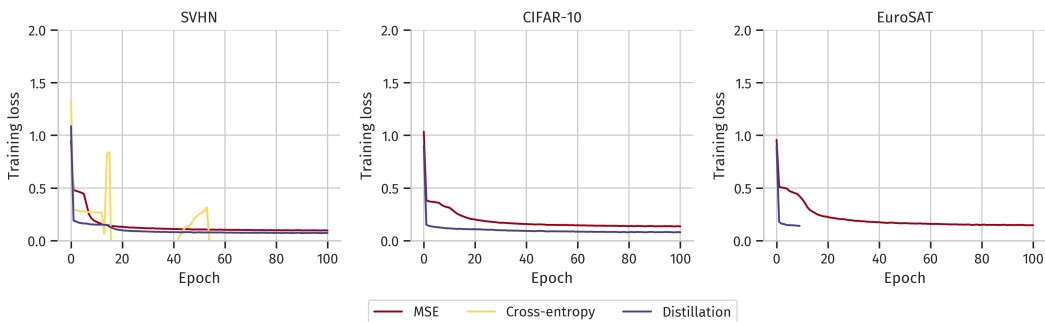

Figure 6: Training loss of hyper-representation models that are trained with all three of contrastive $\mathcal{L}_C$, structural $\mathcal{L}_S$ and behavioral $\mathcal{L}_B$ elements, but where we vary how the behavioral loss is computed. We note that only the MSE loss yields stable results across all model zoos.

MSE loss yields already very satisfying results, we decided to focus solely on the MSE loss in this paper.

### D.3 USING THE RIGHT QUERIES IS ESSENTIAL

While computing the structural loss is a straightforward MSE between the weights of the original model and its reconstruction, comparing their behavior requires using some data, which we have named the *queries* $X = \{x_i\}_{i=1}^n$. As discussed in Section 3, this raises the question of what data points to use as queries. In this context, we name dataset from which we sample the $n$ queries the *query set*. In this Appendix, we discuss the choice of the query set.

Let a model $f_{\theta_j}$ be trained over some training dataset $(X, y)$. First, since the behavioral loss only compares the outputs of $f_{\theta_j}$ and its reconstruction $f_{\hat{\theta}_j}$, we do not need $y$ and can discard it. We therefore focus on $X$, and its distribution $p(X)$. Indeed, since the input domain $\mathcal{X}$ can be very large, e.g., all pixel values in a $32 \times 32 \times 3$ image, we have guarantees to sample from the subdomain of interest only if we are restricted to a very small subset, e.g., natural images. Similarly, the behavior of $f_{\theta_j}$ is well defined only where $p(X)$ has high probability, as it is the domain it has been trained on. The existence of adversarial examples (Goodfellow et al., 2014) shows that even in the close vicinity of $p(X)$, the behavior of $f_{\theta_j}$ is ill-defined. It follows that for our behavioral loss to be effective at reconstructing NNs of similar performance, it needs to be computed as close as possible to $p(X)$.

We experimentally validate this hypothesis. The setup is the same as the one described in Section 4.1. We train hyper-representation AEs on all three of our model zoos, using all three of the contrastive $\mathcal{L}_C$, structural $\mathcal{L}_S$ and behavioral $\mathcal{L}_B$ losses. Here, however, we vary the *query set* used when computing the behavioral loss during training. The baseline, which we name "*Zoo trainset*", is the same setup as used in Section 4, where the queries are samples from the training set of the corresponding model zoo. The first variation consists in using natural images, but from a different dataset. This could correspond to a setup where the hyper-representation model does not have access to the zoo's training set and uses some default dataset instead. To that end, we use the STL-10 dataset (Coates et al., 2011), resized to $32 \times 32 \times 3$, which we name "*STL-10_32*". Finally, we also test using data points sampled uniformly from $\mathcal{X}$, i.e., uniform noise; we name this query set "*Random*".

We show the results for the discriminative downstream tasks in Figure 7. As hypothesized, we see large gaps in performance depending on the query set used. In all cases, when using the *Random* query set the performance is low. For both SVHN and EuroSAT, we also see a large gap in performance between *Zoo trainset* and *STL-10_32*. For CIFAR-10, however, the performance when using *STL-10_32* is very good, even surpassing *Zoo trainset*. Indeed, STL-10 is a dataset that also contains natural images, and uses similar classes as CIFAR-10. The distributions of SVHN and EuroSAT, respectively house numbers and satellite imaging, are farther in distribution from STL-10.

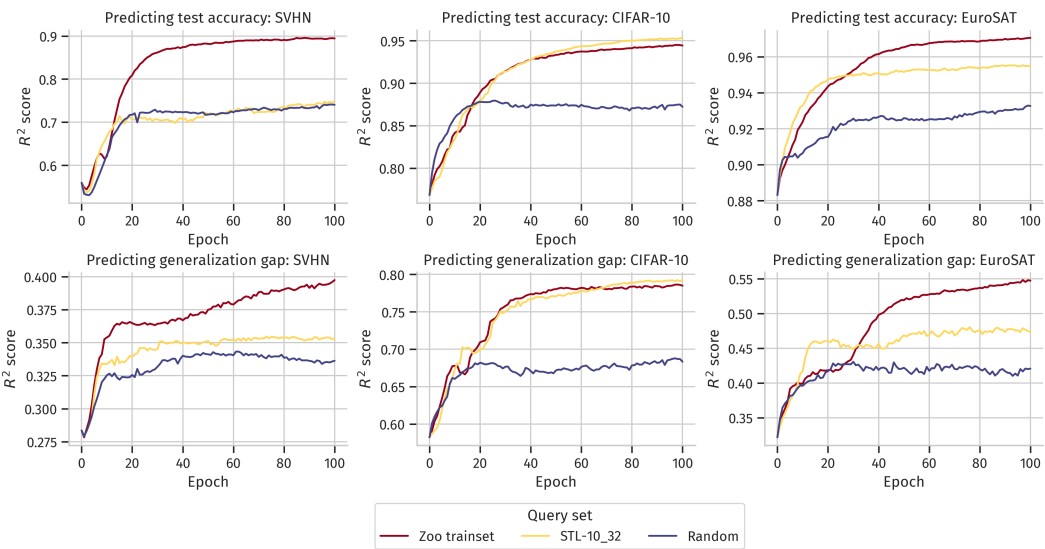

Figure 7: Comparison of the performance of the discriminative downstream tasks by training epoch of `SANE`, compared for different query sets. *Random* queries yield low performance, and using the *Zoo trainset* yields highest performance for SVHN and EuroSAT. For CIFAR-10, *Zoo trainset* and *STL-10_32* show a similar level of performance.

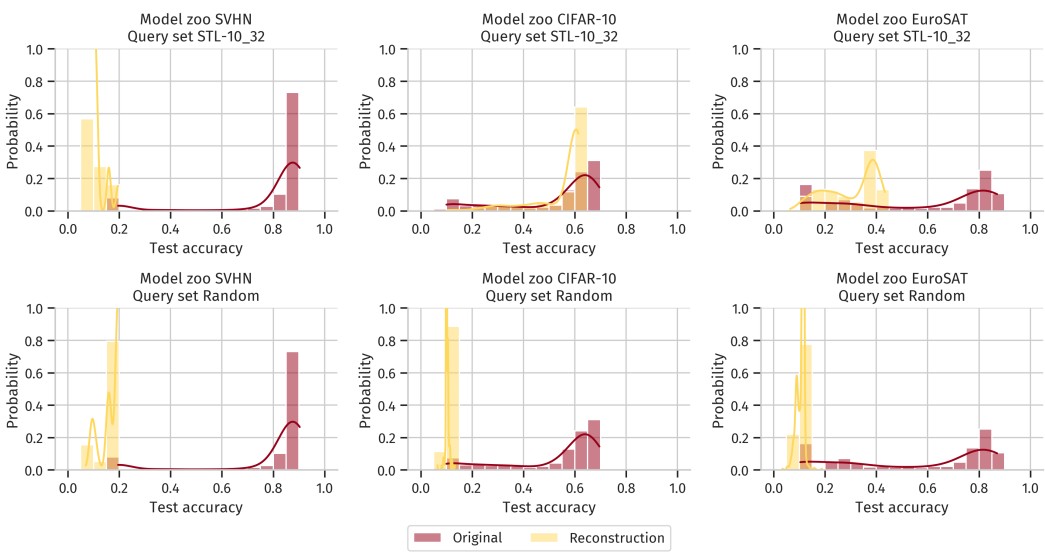

Figure 8: Evaluation of the model reconstructions, shown as distributions of the test accuracy of different models, depending on whether they are part of the original model zoo or reconstructions. We represent variations in the query set in the rows, where *STL-10_32* and *Random* are represented. Models reconstructed using *Random* queries all perform close to random guessing, and so do SVHN models reconstructed with *STL-10_32* queries. EuroSAT models reconstructed with *STL-10_32* queries perform a little better than random guessing but a lot worse than original models. CIFAR-10 models reconstructed with *STL-10_32* tend to match the distribution of the original models, except the most performant ones, where they fail to perform as well.

We further show the results for the reconstructive downstream tasks in Figure 8. When using the *Random* query set, we see a similarly poor performance, with reconstructed models all performing poorly, very close to random guessing. For the SVHN zoo, using *STL-10_32* performs similarly poorly. It performs slightly better for EuroSAT. When considering CIFAR-10, reconstruction using the *STL-10_32* query set performs relatively well, but reconstructed models still fail to match the very best models in the zoo in terms of performance.

These results validate our hypothesis that the choice of a right query set is of paramount importance. That is a limitation for the behavioral loss, as contrary to the structural one, it is not data free, and even requires data from the zoo's training set to perform best. Results on CIFAR-10 and STL-10 are, however, encouraging: using a query set that is not the zoo's training set, but that is close enough in distribution, can still yield very satisfying levels of performance. This opens the door for engineering comprehensive query sets for cases where the training set of the models in the zoo is not available.

## D.4 Behavioral Loss Remains Relevant with Limited Computing Resources

As discussed in Appendix B, training hyper-representation models with a behavioral loss takes around twice as much computing time as with only a structural loss. There, we concluded that this was an acceptable trade-off with regard to the increased performance the behavioral loss brings. This raises, however, the question of the validity of the behavioral loss in a compute-constrained setting. In this Section, we evaluate the performance of hyper-representation AEs trained with either the structural or both the structural and behavioral loss, but for a similar computing budget.

To that end, we consider the experiment setup as defined in Section 4.1. We compare the hyper-representation model trained with the contrastive $\mathcal{L}_C$ and structural $\mathcal{L}_S$ losses with the one trained with all three, i.e., the one that also includes the behavioral loss $\mathcal{L}_B$. The difference is, we take the hyper-representation $\mathcal{L}_C + \mathcal{L}_S$ after 100 epochs of training, and compare it to the $\mathcal{L}_C + \mathcal{L}_S + \mathcal{L}_B$ after only 50 epochs; the computing budget allocated to both is therefore comparable.

When comparing the discriminative downstream tasks' performance in Figure 5, we clearly see that the performance of the hyper-representation trained with $\mathcal{L}_C + \mathcal{L}_S + \mathcal{L}_B$ at 50 epochs is in all cases higher than that of the one trained on $\mathcal{L}_C + \mathcal{L}_S$ at 100 epochs. While the performance of the latter seems to increase more rapidly, it reaches a plateau relatively early in the training process; the former, however, keeps improving steadily.

We further investigate the reconstructive downstream tasks in Figure 9. There, we see that even with less training epochs, the models reconstructed using the behavioral loss $\mathcal{L}_B$ reach higher levels of performance, comparable to that of the models from the zoo. We note, however, that for the EuroSAT zoo, we fail to reconstruct models as performant as the best models in the zoo: this shows that the AE could still benefit from further training. In general, however, these results indicate that for a fixed computing budget, using both the structural $\mathcal{L}_S$ and behavioral $\mathcal{L}_B$ losses yields better performance than using $\mathcal{L}_S$ only with more training epochs.

## D.5 Combining Structure and Behavior Works on Larger Architectures

In this Appendix, we explore whether our findings generalize to a larger CNN architecture. To do so, we build a new model zoo using the same hyperparameters as those described in Appendix A but based on the LeNet-5 (LeCun et al., 1998) architecture, for the CIFAR-10 dataset. This architecture has $62,006$ parameters, or roughly 6 times as many parameters compared to our other zoos. We show results in Figure 10, which validate that using $\mathcal{L}_C + \mathcal{L}_S + \mathcal{L}_B$ outperforms the baseline ($\mathcal{L}_C + \mathcal{L}_S$) for reconstructive downstream tasks.

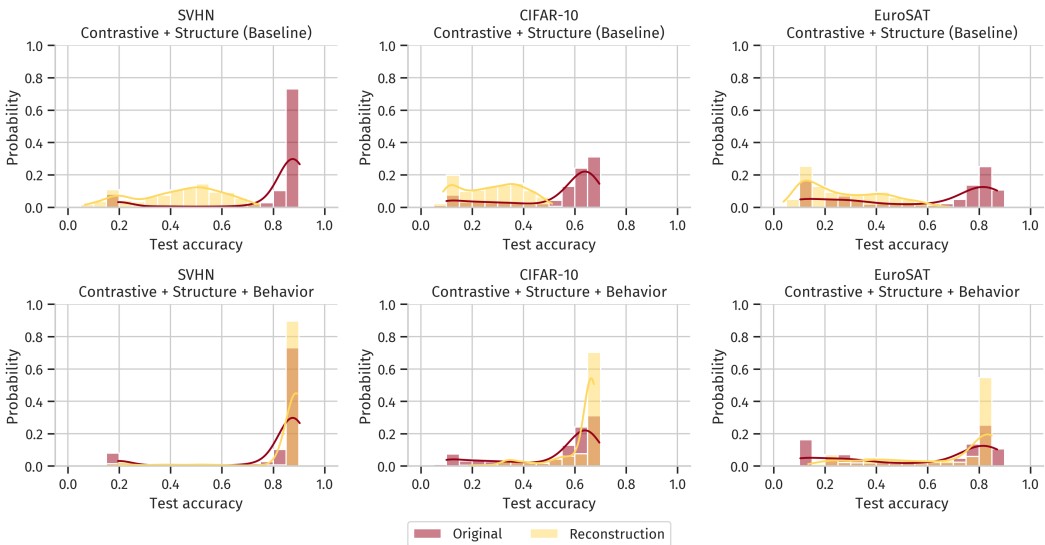

Figure 9: Evaluation reconstructive downstream tasks, for a comparable computing budget where $\mathcal{L}_C + \mathcal{L}_S$ is evaluated after 100 epochs and $\mathcal{L}_C + \mathcal{L}_S + \mathcal{L}_B$ is evaluated after only 50 epochs. Even with half the number of training epochs, the AE using the behavioral loss outperforms the one that does not use it. We see, however, that for the EuroSAT zoo, we fail to reconstruct the very best models as performant as they are: the AE could benefit from further training.

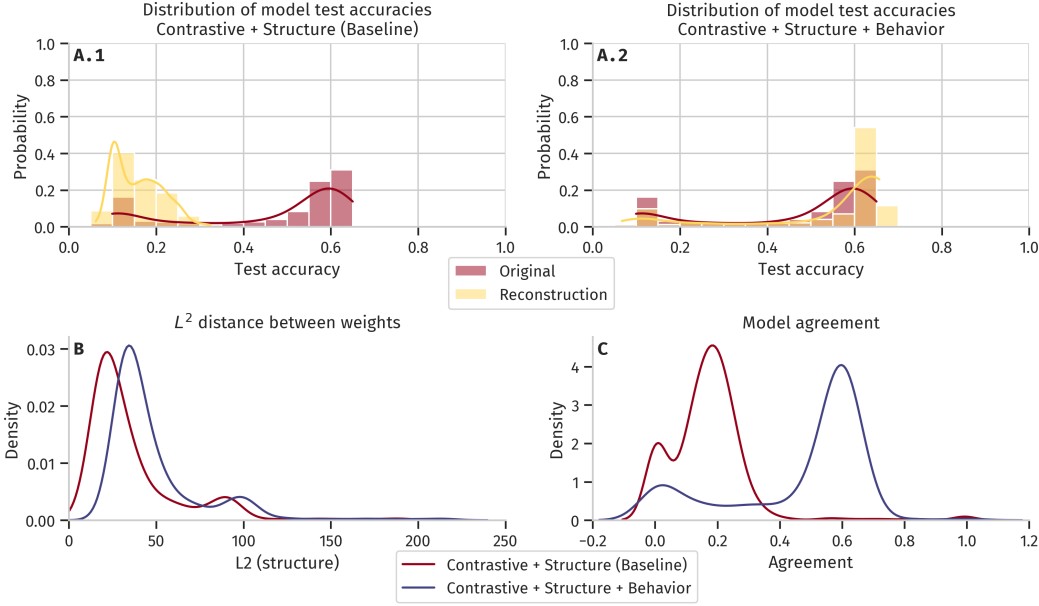

Figure 10: Evaluation of reconstructive downstream tasks, on a larger LeNet-5 architecture, trained on CIFAR-10. Panels A.1 and A.2 show the distribution of test accuracies of models and their reconstruction. They show that the distribution of reconstructed models' performance is much closer to that of the originals' when using both $\mathcal{L}_S$ and $\mathcal{L}_B$. Panel B shows the pairwise $L^2$ distances between the weights of models and their respective reconstruction; we note a similar distribution with slightly lower distances when using $\mathcal{L}_S$ only. Panel C shows the pairwise model agreement between models and their respective reconstruction, demonstrating a higher level of agreement when using both $\mathcal{L}_S$ and $\mathcal{L}_B$.

## D.6 SELECTING THE WEIGHT OF THE STRUCTURAL LOSS

In this Appendix, we describe how we selected the hyperparameter $\beta$, which is used to control the weight of the structural loss $\mathcal{L}_S$ and the behavioral loss $\mathcal{L}_B$. To do so, we used the validation set of our model zoo, and evaluated the discriminative and reconstructive downstream tasks on it using different values of hyperparameter $\beta$.

We show the results for the discriminative downstream tasks in Table 7. There, a value of $\beta = 0.1$ shows best performance across all datasets, except when predicting the generalization gap on CIFAR-10. Differences between the performance of the downstream tasks are larger when considering the SVHN dataset than for the other two. For these reasons, with regard to the discriminative downstream tasks, we select a value of $\beta = 0.1$.

We show the results for the reconstructive downstream tasks in Table 8. As can be expected, a larger $\beta$, which is linked with a larger weight for the structural loss $\mathcal{L}_S$, generally leads to lower $L^2$ distance between model weights. When considering model agreement, however, a value of $\beta = 0.1$ seems to perform best across all datasets. Since this value performs best for both discriminative and reconstructive downstream tasks, we select it for the rest of our experiments.

Table 7: Discriminative downstream tasks performance on the validation set, for various values of hyperparameter $\beta$. We predict the test accuracy and generalization gap of our models based on their latent representation, using a linear probe. We give the $R^2$ score for predictions on the held-out test split. For the SVHN dataset, using $\beta = 0.1$ clearly outperforms all other options. When predicting model test accuracy, it is also the best option, by a small margin. Results for predicting the generalization gap are more inconclusive for CIFAR-10 and EuroSAT, but differences in performance remain relatively small.

| | **TEST ACCURACY** | | | **GENERALIZATION GAP** | | |
| $\beta$ | SVHN | CIFAR-10 | EuroSAT | SVHN | CIFAR-10 | EuroSAT |
|---|---|---|---|---|---|---|
| 0.0 | 0.725 | 0.932 | 0.940 | 0.338 | 0.799 | **0.622** |
| 0.1 | **0.890** | **0.964** | **0.963** | **0.429** | 0.788 | **0.622** |
| 0.2 | 0.667 | 0.958 | 0.957 | 0.295 | 0.789 | 0.576 |
| 0.3 | 0.657 | 0.959 | 0.962 | 0.290 | 0.776 | 0.557 |
| 0.4 | 0.668 | 0.958 | 0.948 | 0.324 | 0.806 | 0.578 |
| 0.5 | 0.701 | 0.941 | 0.943 | 0.337 | **0.807** | 0.536 |

Table 8: Reconstructive downstream tasks performance on the validation set, for various values of hyperparameter $\beta$. We evaluate structural reconstruction with the average $L^2$ distances between the weights of test split models and their reconstructions. We evaluate behavioral reconstruction with the average classification agreement between test split models and their reconstructions. Standard deviation is indicated between parentheses. When considering structural distance, a larger $\beta$ seems linked with a lower distance. When considering model agreement, a value of $\beta = 0.1$ performs best across all three datasets.

| | **STRUCTURE ($L^2$ DISTANCE)** | | | **BEHAVIOR (AGREEMENT)** | | |
| $\beta$ | SVHN | CIFAR-10 | EuroSAT | SVHN | CIFAR-10 | EuroSAT |
|---|---|---|---|---|---|---|
| 0.0 | 56.0 (±6) | 61.0 (±9) | 64.2 (±16) | 19.6% (±1%) | 10.5% (±4%) | 10.8% (±4%) |
| 0.1 | 27.1 (±9) | 30.0 (±12) | 40.6 (±20) | **82.1% (±18%)** | **54.5% (±13%)** | **66.6% (±23%)** |
| 0.2 | 26.7 (±8) | 28.3 (±12) | 39.5 (±20) | 64.9% (±18%) | 53.5% (±16%) | 64.9% (±25%) |
| 0.3 | 26.3 (±8) | 27.9 (±11) | 37.4 (±18) | 65.2% (±18%) | 51.4% (±18%) | 63.9% (±26%) |
| 0.4 | **25.5 (±8)** | **27.5 (±10)** | 37.0 (±18) | 65.0% (±18%) | 50.6% (±18%) | 62.8% (±26%) |
| 0.5 | 25.6 (±8) | 28.0 (±10) | **35.9 (±18)** | 64.9% (±18%) | 47.5% (±16%) | 61.4% (±26%) |

