# OpenReview forum: "Structure and Behavior in Weight Space Representation Learning"
_ICLR.cc/2025/Conference — Submitted to ICLR 2025_

### Official Review · Reviewer_iCWh · 2024-10-23

**Soundness:** 3
**Presentation:** 3
**Contribution:** 2
**Rating:** 5
**Confidence:** 4

**Summary:**

The paper discusses autoencoding of weight-spaces. In this field of research,
weights of deep models are viewed as an object (input and output) of learning systems.
Specifically, they look to improve the function of models encoded using weight-space AEs,
by adding what they call, a "behavioral" loss, designed to improve network function.
That loss is simply an L2 loss over the output of the model with the reconstructed weights.

The paper runs meaningful experiments largely following the standards in the field. On three model populations "zoos", they show that adding the new loss improves the AE, in terms of generating and reconstructing more faithful weights.

My recommendation is borderline reject, because of the limited technical novelty and conceptual contribution, which means the audience for this paper would be very narrow. I am willing to be convinced otherwise.

**Strengths:**

-- The paper tackles an important problem in the growing field of weight-space modality.

-- The paper contains a large set of evaluation experiments for generation and reconstruction of weights in three model zoos.

**Weaknesses:**

W1:  The idea is very straight forward, and the technical contribution is quite limited. Basically, they run experiments like those done previously in the field, while adding the new L2 functional loss. I am not aware that others have done this for weights autoencoders, but using "behavior" (out put of model parameterized by weights) to select weights has been in other context. For example a loss over the function parametrized by the weights is the standard way to train hypernetworks.

W2:  The PCA analysis of Figure 1 has two issues. (1) It is suspicious (see questions below) and (2) it is not clear how it connects to the contribution of the paper, since the analysis is not used as far as I can tell to motivate or guide the method or the experiments.

W3:  Every evaluation of the behavior loss involves 500 inference steps through the classifier model (250 samples x 2 models). How does this added compute compares with the total compute for training the AE? If it increases the overall compute required to train the AE, it should be explicitly stated.

W4: Definition of behavioral loss. For classifiers, L2 is not a great choice. One should use the same loss that the models from the zoo cis trained with, . e.g. cross entropy for classification etc.


Specific and minor comments and suggestions

-- Define key terms like "structural reconstruction error" and "model zoo" before using them. Add concrete definitions of all losses.
    The paper should be made readable to researchers outside the weight-space community.

-- I found Fig 3 totally confusing. Is there anything interesting insight to take from these histograms tat cannot be conveyed in a simpler way? Distributions are quite unimodal so a mean+std table would have been much simpler and easy to understand.

-- Related work on generation of NN weights. It would make sense to mention Hypernetworks.

**Questions:**

Q1: Figure 1 looks highly suspicious, and many details are skipped. How come it is useful to add "Bottom" eigenvectors, which capture directions in space that have little to no information about the weights? How does the eigen spectrum look like? and what is the dependency of explained variance on the number of principal components? What is "reconstruction accuracy" in this figure? (the text said you measured test-set accuracy).
What was the size of the training data and weight-vector dimension? same as in line 254? I suspect that the covariance matrix was low rank, and that the correct x-axis in the figure is "fraction of of eigen vectors taken", not "explained variance". That may explain the totally flat curves on the middle and right panels.   And how comes the random accuracy in SVHN is 20%?

Q2:  Not stated clearly how the loss weights were selected. Validation sets?

---

> ### Author Response · Authors · 2024-11-20
> **Response to reviewer iCWh (1)**
>
> We thank reviewer iCWh for their review, and will address the weaknesses they note and the questions they raise in this response.
>
> > W1: The idea is very straight forward, and the technical contribution is quite limited. Basically, they run experiments like those done previously in the field, while adding the new L2 functional loss. I am not aware that others have done this for weights autoencoders, but using "behavior" (out put of model parameterized by weights) to select weights has been in other context. For example a loss over the function parametrized by the weights is the standard way to train hypernetworks.
>
> We agree with the reviewer that using a probing-based loss in itself is not a novel thing, and have added a relevant paragraph to our related work Section. As mentioned in the response to all reviewers, however, the scope of our contribution goes beyond that. By motivating why we use it, what issues it solves and how it works, we hope to bring more explanatory elements in understanding weight space self-supervised learning, and to highlight the importance of the structure-behaviour synergy for future works in the domain. Our experiments using the SANE [1] framework aim at validating these findings.
>
> > W2: The PCA analysis of Figure 1 has two issues. (1) It is suspicious (see questions below) and (2) it is not clear how it connects to the contribution of the paper, since the analysis is not used as far as I can tell to motivate or guide the method or the experiments.
>
> > Q1: Figure 1 looks highly suspicious, and many details are skipped. How come it is useful to add "Bottom" eigenvectors, which capture directions in space that have little to no information about the weights? How does the eigen spectrum look like? and what is the dependency of explained variance on the number of principal components? What is "reconstruction accuracy" in this figure? (the text said you measured test-set accuracy).
> What was the size of the training data and weight-vector dimension? same as in line 254? I suspect that the covariance matrix was low rank, and that the correct x-axis in the figure is "fraction of of eigen vectors taken", not "explained variance". That may explain the totally flat curves on the middle and right panels. And how comes the random accuracy in SVHN is 20%?
>
> Figure 1 and the related experiment are highly related to that from Balestriero and LeCun [2]. In their work on autoencoders for computer vision, they show that the features most useful for classification of images lie with the lowest eigenvalues, although the features useful for reconstruction lie with the largest eigenvalues. In a similar fashion, we show that in weight space, one needs both features that lie with the “Top” and “Bottom” eigenvalues to reconstruct high-performing models.
>
> Models are trained on the training sets of the corresponding datasets, and tested on the test sets. Weight vectors are flattened model weights, therefore length $10,853$. As the reviewer correctly guesses, the resulting covariance matrix is low rank. For this reason, we specifically target “Top” or “Bottom” eigenvalues that explain a certain amount of variance, and not a certain proportion of the number of eigenvalues. It follows that the bottom 10\% explained variance contains many more eigenvalues than the top 10\%, but they explain the same amount of variance. Finally, the first point in all plots sits at 10\% explained variance, not 0\%; the corresponding model is not random but reconstructed from eigenvalues that explain 10\% of the variance, and therefore performs better than random guessing. We hope these details convince the reviewer of the soundness of Figure 1. We further add that the supplementary material contains the corresponding code and can be verified.
>
> Regarding point (2), the goal with the Figure 1 experiment is to highlight that structural reconstruction, which naturally focuses on the features spanned by the top eigenvectors, will miss out some features that are essential for reconstructing high-performing models. As such, we motivate the need for an additional loss element that focuses on such features. Such a loss needs to be used in conjunction with the structural loss, since we also show that the features spanned by the top eigenvectors remain relevant. In Section 3, we derive our behavioural loss, and describe how it will focus on different features than the structural loss. In Section 4, we experimentally validate that with our new loss, we have indeed managed to target these missing features through the addition of the behavioural loss element. Whether that loss specifically targeted features spanned by the bottom eigenvectors remains however elusive, as the eigenvectors of reconstructed models are mostly misaligned with those of the original ones. Our results nevertheless empirically show that relevant features have been learned, since we do manage to reconstruct high-performing models.

---

> > ### Author Response · Authors · 2024-11-20
> > **Response to reviewer iCWh (2)**
> >
> > > W3: Every evaluation of the behavior loss involves 500 inference steps through the classifier model (250 samples x 2 models). How does this added compute compares with the total compute for training the AE? If it increases the overall compute required to train the AE, it should be explicitly stated.
> >
> > As mentioned in the Limitations section, there is indeed an increase in computational cost. We have further explored that matter in Appendix D.4, comparing experiments where we approximately control for training time. In particular, we show that a model trained with both structure and behaviour still outperforms a model trained for more epochs with the structural loss only.
> >
> > > W4: Definition of behavioral loss. For classifiers, L2 is not a great choice. One should use the same loss that the models from the zoo cis trained with, . e.g. cross entropy for classification etc.
> >
> > In this work, we attempt to match the behaviour of a model and its reconstruction, irrespective of the task they solve and the correctness of their predictions. As such, using the $L^2$ distance as a measure of the differences between their predictions is a valid approach: we simply perform a regression on the original model’s behaviour. We have nevertheless explored the use of other loss functions in Appendix D.2, and have shown that using a cross-entropy loss is less stable than using a simpler $L^2$ loss. What is more, when using a $L^2$ loss, our framework needs no adaptation to work on regression or reconstruction models, which do not output class probabilities.
> >
> > > Define key terms like "structural reconstruction error" and "model zoo" before using them. Add concrete definitions of all losses. The paper should be made readable to researchers outside the weight-space community.
> >
> > We agree that the concept of a model zoo should be defined for readers not familiar with the field: we have added a short definition in our Section 4.1. Regarding the loss functions, the structural loss is defined in Equation (3) and the behavioural loss in Equation (2). The contrastive loss is a standard NTXent loss, which is mentioned and adequately cited in Section 4.1.
> >
> > > I found Fig 3 totally confusing. Is there anything interesting insight to take from these histograms tat cannot be conveyed in a simpler way? Distributions are quite unimodal so a mean+std table would have been much simpler and easy to understand.
> >
> > We disagree with the reviewer that these distributions are unimodal, all distributions show models in the low test accuracy regime, and corresponding reconstructions. For this reason, we do not believe that the mean and standard deviation of the distributions would convey the same amount of information as Figure 3 does. This bi-modality appears even more clearly in the new experiments we introduced in the Appendix D.5 with LeNet-5 models. We provide numbers for comparison purposes, in particular the pairwise similarity metrics between models and their reconstructions, in Table 5. That Table includes relevant averages and standard deviations.
> >
> > > Related work on generation of NN weights. It would make sense to mention Hypernetworks.
> >
> > We do mention HyperNetworks in our related work Section, but we chose to do so in the “Weight Space Representation Learning” paragraph.
> >
> > > Q2: Not stated clearly how the loss weights were selected. Validation sets?
> >
> > It is indeed an oversight on our side not to have included those experiments. Hyperparameter $\gamma$ comes from the original SANE [1] implementation. Hyperparameter $\beta$ has indeed been selected using validation sets. As mentioned in the response to all reviewers, we have added those experiments on the validation sets as Appendix D.6.
> >
> > —
> >
> > [1] Schürholt et al., "Towards Scalable and Versatile Weight Space Learning.", ICML 2024.
> >
> > [2] Balestriero and LeCun, “Learning by reconstruction produces uninformative features for perception”, ICML 2024.

---

> > > ### Comment · Reviewer_iCWh · 2024-11-25
> > > **Discussion of rebuttal**
> > >
> > > Thank you for your detailed response to my comments. Specific concerns, like my misunderstanding of Figure 1 were clarified.
> > >
> > > I appreciate the careful work that authors have invested into this paper. Unfortunately, I feel that the level of conceptual advance and significance does not reach the bar needed for an ICLR paper. I cannot recommend the paper for accepetance.

---

### Official Review · Reviewer_KkRG · 2024-10-24

**Soundness:** 2
**Presentation:** 2
**Contribution:** 2
**Rating:** 3
**Confidence:** 4

**Summary:**

The paper considers the problem of constructing weight space encoders, i.e., learning an Auto-Encoder (AE) on the weights of trained NNs (for example, image classifiers). Previous works mainly considered two loss functions: a “structural” reconstruction loss, e.g., MSE between the reconstructed and input weights, and a contrastive loss. The authors propose augmenting the two losses with an additional loss, which they term “behavioral”: This loss is the MSE between the outputs of the original and reconstructed NN, evaluated on a set of inputs.

**Strengths:**

1. The paper is generally well-structured and easy to follow.
1. The empirical evidence for the effectiveness of behavioral loss is strong.

**Weaknesses:**

My main concern is the paper's limited novelty and contribution. The paper's main claimed novel contribution is the inclusion of behavioral loss.
1. The proposed approach generally follows the SANE [1] paper, model arch, and experimental setup, with the only difference in the loss function. While the results presented show improvements over SANE, the contribution of the paper is limited.
1. Previous works for weight space learning used similar loss functions or probing-based features. At the very least, this deserves some reference and discussion by the authors. A similar loss function, although not identical (and not always in the context of AE), was proposed in e.g. [2, 3, 4, 5]. The idea is to evaluate the reconstructed model on a set of inputs and compute the loss w.r.t ground truth labels. This is very close to the proposed approach, with the difference being that the proposed approach uses pseudo labels instead of GT labels. However, this is almost identical in the context of INRs [2, 3].
Another relevant literature that is not discussed, is the inclusion of probing-based features, which evaluate the original function on (fixed or learned) inputs to provide additional information on the input net and functional behavior [6, 7].
1. Experiments are performed on small-scale NNs and datasets. It is not clear how this generalizes to real-world large-scale setups.
1. Please also include some measure of variability (e.g., standard derivation) in Table 1.
1. The results of Section 2 suggest that it is important to learn to match eigenvectors corresponding to both large and small eigenvalues, while that of mid-sized eigenvalues is less important. This provides a good motivation for augmenting the structural only loss, however:
    - It is not clear to me that the proposed loss achieves this specific goal.
    - Also, it would be beneficial to show some empirical evidence that including behavioral loss is improving the reconstruction of subspaces spanned by both the top and bottom eigenvectors.
1. As also noted by the authors, the proposed approach increases the computational cost. This increase may be too expensive for large-scale models (which are not included in this work).

References:

[1] Schürholt et al. "Towards Scalable and Versatile Weight Space Learning." ICML 2024.

[2] Luigi et al. "Deep Learning on Implicit Neural Representations of Shapes." ICLR 2023.

[3] Navon et al. “Equivariant architectures for learning in deep weight spaces,” ICML 2023.

[4] Zhou et al. "Permutation Equivariant Neural Functionals." NeurIPS 2023.

[5] Navon et al. “Equivariant Deep Weight Space Alignment,” ICML 2024.

[6] Kofinas et al. "Graph Neural Networks for Learning Equivariant Representations of Neural Networks." ICLR 2024.

[7] Herrmann et al. "Learning useful representations of recurrent neural network weight matrices.", 2024.

**Questions:**

1. From Figure 3 and as pointed out by the authors, it appears that the behavior of reconstructed models does not necessarily reproduce that of the original model, rather, it is biased towards higher accuracies. Why do you think that is the case? These results suggest that the reconstruction of low performing models is not good, even with the added behavioral loss.
1. In Table 2, selecting the Max statistics seems too convenient. Using the mean or median or Min, for example, would have shown significant differences between the original and generated models.

---

> ### Author Response · Authors · 2024-11-20
> **Response to reviewer KkRG (1)**
>
> We would like to thank reviewer KkRG for their review, and will try to address the weaknesses and questions they raise.
>
> > The proposed approach generally follows the SANE [1] paper, model arch, and experimental setup, with the only difference in the loss function. While the results presented show improvements over SANE, the contribution of the paper is limited.
>
> As mentioned in the response to all reviewers, we do not believe our main contribution lies in adding a behavioural loss into the SANE framework. With our submission, we identify a particular issue in structural representation learning, derive and propose a solution, which highlights synergies between structural and behavioural signals in self-supervised weight space learning. The experiments we run using SANE [1] are used to validate our findings through the strength of the results when combining structural and behavioural loss elements.
>
> > Previous works for weight space learning used similar loss functions or probing-based features. At the very least, this deserves some reference and discussion by the authors. A similar loss function, although not identical (and not always in the context of AE), was proposed in e.g. [2, 3, 4, 5]. The idea is to evaluate the reconstructed model on a set of inputs and compute the loss w.r.t ground truth labels. This is very close to the proposed approach, with the difference being that the proposed approach uses pseudo labels instead of GT labels. However, this is almost identical in the context of INRs [2, 3]. Another relevant literature that is not discussed, is the inclusion of probing-based features, which evaluate the original function on (fixed or learned) inputs to provide additional information on the input net and functional behavior [6, 7].
>
> We would like to thank the reviewer for the literature that they provide. We discuss how these works differ from ours here, and have also added a relevant paragraph in the related work Section of our manuscript.
>
> De Luigi et al. [2] do use a probing-based loss in the context of autoencoders, but without incorporating a structural element. In addition, the queries are part of the inputs of the decoder, and they directly predict the true labels, whereas we favour reconstruction. Additionally, [2] performs reconstruction of the shapes represented by INRs, not INRs themselves, making direct comparison to our work difficult. To the best of our understanding, [3] mostly leverages structural signals in a supervised way, or contrastive signals for self-supervised learning. It uses a probing-based signal for domain adaptation using true labels on a corrupted dataset, but it seems limited to this use-case. [7] show high performance levels from using probing-based mechanisms to analyse recurrent neural networks, but we note that they use their decoders as an emulator of the function represented by the original model, whereas we directly reconstruct the original model. As mentioned in the response to reviewer fxSa, the learning setup in [5] is the closest to ours, as the authors leverage a mix of supervised, structural and behavioural signals, which shows great performance. Their probing-based loss, however, requires true labels and does not compare the behaviour of two models in the same way our behavioural loss does. With our contribution, we want to demonstrate that combining structural and behavioural signals plays a part in this, and to show the importance of that dual signal for the development of future weight-space methods.
>
> > Experiments are performed on small-scale NNs and datasets. It is not clear how this generalizes to real-world large-scale setups.
>
> As explained above, in this work we aim to understand a fundamental issue in structural weight space learning. To that end, we evaluate failure modes and systematically develop an approach to avoid that failure mode. Our experiments are validations of that.
>
> We acknowledge, however, that a more extensive set of architectures could further strengthen our claim. We address this point in more detail in the response to all reviewers, where we explain architectural challenges in scaling up the models we work on. In order to validate our findings on larger architectures, we have added a new Appendix D.5 where we run experiments on a new model zoo of LeNet-5 models trained on CIFAR-10, which are around 6 times larger that the models in our other zoos.
>
> > Please also include some measure of variability (e.g., standard derivation) in Table 1.
>
> The results presented in Table 1 are computed as the $R^2$ score over the whole test set. It follows that there is no direct standard deviation measure to provide. Where applicable, such as in Table 5 and 6, standard deviation metrics are provided.

---

> > ### Author Response · Authors · 2024-11-20
> > **Response to reviewer KkRG (2)**
> >
> > >The results of Section 2 suggest that it is important to learn to match eigenvectors corresponding to both large and small eigenvalues, while that of mid-sized eigenvalues is less important. This provides a good motivation for augmenting the structural only loss, however:
> > > 1. It is not clear to me that the proposed loss achieves this specific goal.
> > > 2. Also, it would be beneficial to show some empirical evidence that including behavioral loss is improving the reconstruction of subspaces spanned by both the top and bottom eigenvectors.
> >
> > With Section 2, we mainly want to highlight that structural reconstruction, which naturally focuses on the features spanned by the top eigenvectors, will miss out some features that are essential for reconstructing high-performing models. As such, we motivate the need for an additional loss element that focuses on such features. Such a loss needs to be used in conjunction with the structural loss, since we also show that the features spanned by the top eigenvectors remain relevant.
> >
> > In Section 3, we derive our behavioural loss, and describe how it will focus on different features than the structural loss. In Section 4, we experimentally validate that with our new loss, we have indeed managed to target these missing features through the addition of the behavioural loss element. Whether that loss specifically targeted features spanned by the bottom eigenvectors remains however elusive, as the eigenvectors of reconstructed models are mostly misaligned with those of the original ones. Our results nevertheless empirically show that relevant features have been learned, since we do manage to reconstruct high-performing models.
> >
> > > As also noted by the authors, the proposed approach increases the computational cost. This increase may be too expensive for large-scale models (which are not included in this work).
> >
> > This is true, and we note this in our Limitations Section. In Appendix D.4, we nevertheless show that even when using comparable computing resources, using both structure and behaviour still performs better than using only structure and running more epochs. It should therefore be possible to use a combination of structure and behaviour with minimal computational overhead.
> >
> > > From Figure 3 and as pointed out by the authors, it appears that the behavior of reconstructed models does not necessarily reproduce that of the original model, rather, it is biased towards higher accuracies. Why do you think that is the case? These results suggest that the reconstruction of low performing models is not good, even with the added behavioral loss.
> >
> > We agree that Figure 3 does not show this in a very clear way. In the Appendix, Table 5 shows pairwise structural and behavioural distances between models and their reconstructions, and shows that adding the behavioural loss makes reconstructed models significantly more behaviourally similar to the original ones. In addition, the new experiments we ran on the larger architectures in Appendix D.5 show a stronger bi-modal model performance in the model zoo, which is adequately matched by the bi-modal distribution of corresponding reconstructed models. There is still a bias towards higher accuracies: we hypothesise it is linked with the higher number of high-performing models compared to low-performing ones, but lack strong evidence to back any definitive claim in that regard.
> >
> > > In Table 2, selecting the Max statistics seems too convenient. Using the mean or median or Min, for example, would have shown significant differences between the original and generated models.
> >
> > A major caveat of the existing structure-only based approach, as shown in [1, 8], is the impossibility to reconstruct or generate models with a good enough fidelity that they show high test accuracy. In Table 2, by showing the Maximum achieved accuracy, we effectively demonstrate that leveraging the structure-behaviour synergy effectively overcomes that limitation. In Table 5, which is in the Appendix, we provide more statistics, specifically the pairwise structural and behavioural similarities of models and their reconstructions. These results show that using both structure and behaviour significantly improves reconstruction behavioural similarity in the general case, not only for the best-performing model(s). Finally, since the distribution of model accuracies is not unimodal, we do not think that adding the mean test performance of the model population would be very relevant.
> >
> > —
> >
> > [1-7] are identical to the reviewer’s reference numbers, for consistency purposes.
> >
> > [8] Schürholt et al., “Hyper-representations as generative models: Sampling unseen neural network weights”, NeurIPS 2022

---

> > > ### Comment · Reviewer_KkRG · 2024-11-25
> > > **Response to Authors**
> > >
> > > I would like to thank the authors for their response. I have gone through the reviews and the authors' responses. I believe my main concerns regarding novelty and empirical evaluation remain. I still believe the paper's novel contribution is limited. Hence, I will keep my initial score.

---

### Official Review · Reviewer_fxSa · 2024-10-27

**Soundness:** 3
**Presentation:** 3
**Contribution:** 2
**Rating:** 3
**Confidence:** 5

**Summary:**

The authors introduced a loss function called *behavioral loss*, designed to reduce the performance discrepancy between original neural networks and those reconstructed through a weight-space autoencoder. This behavioral loss minimizes the difference between the outputs of the original and reconstructed networks by reducing the norm between their predictions.


The authors claim the following contributions:
- analyzing the error modes of weight-structure reconstruction.
- Introducing the behavioral loss.
- Demonstrating the improvement in loss performance across various experiments.

**Strengths:**

- The paper is well written and easy to follow.
- The motivation for the proposed solution is well explained and investigated.
- Simple solution.

**Weaknesses:**

- My main concern regarding this work is around its novelty the behavioral loss was already used in [1]. In [1] the authors calculated the loss based on the loss on the downstream task. In addition, there is no novelty (and it was not claimed) on the architecture side since the authors used SANE [2] for the weight space AE. How do the authors differentiate their work from [1]?
- Limited experimental section. Given the fact that this work is mostly empirical, I expect the experimental section to be more extensive both in scale and learning setups. The authors focused on discriminative models and specifically only on small CNNs. There is no diversity in the reconstructed model architecture or the downstream tasks.
- It would be interesting to see how significant are $\beta$ and $\gamma$ hyperparameters in the AE optimization.
- The authors mentioned in the appendix that training with behavioral loss nearly doubles the training duration, which poses a challenge for large models.
-Rather than using a common model zoo [3,4], the authors chose to create their own, which raises some questions.
- The authors did not include confidence intervals in the experimental section. Moreover, in Tables 5 and 6 the authors provide additional results which are inclusive due to high std values. The same should be reported in the main text results.
- The only augmentation used for $\mathcal{L}_C$ is a random permutation, why the authors did not use augmentations from [5,6]? [6] should be cited in this work.

--------------
[1] Equivariant Deep Weight Space Alignment, Navon et al.

[2] Towards Scalable and Versatile Weight Space Learning, Schurholt et al.

[3] Model Zoo: A Dataset of Diverse Populations of Neural Network Models, Schurholt et al.

[4] Eurosat Model Zoo: A Dataset and Benchmark on Populations of Neural Networks and Its Sparsified Model Twins, Honegger et al.

[5] Equivariant Architectures for Learning in Deep Weight Spaces, Navon et al.

[6] Improved Generalization of Weight Space Networks via Augmentations, Shamsian et al.

**Questions:**

- Based on the findings in Section 2, have the authors considered explicitly encouraging hybrid eigenvalue features? For example, they might incorporate a KL divergence loss relative to a predefined Beta distribution concentrated in the high and low values, effectively ignoring the intermediate range.

---

> ### Author Response · Authors · 2024-11-20
> **Response to reviewer fxSa (1)**
>
> We would like to thank reviewer fxSa for their review, and for bringing three very interesting papers by Navon, Shamsian et al. to our attention [1, 5, 6]. We hope to address the weaknesses and questions raised by the reviewer with this response.
>
> > My main concern regarding this work is around its novelty the behavioral loss was already used in [1]. In [1] the authors calculated the loss based on the loss on the downstream task. In addition, there is no novelty (and it was not claimed) on the architecture side since the authors used SANE [2] for the weight space AE. How do the authors differentiate their work from [1]?
>
> The work in [1] is very interesting and parallels can indeed be drawn with our work. To the best of our understanding, the authors use a combination of three losses to train their weight-space alignment model: a supervised loss $L_{supervised}$, an unsupervised loss that minimises weight distance $L_{alignment}$, and finally an unsupervised loss that minimises the model’s loss function on a linear combination of both models’ weights (i.e. the linear mode connectivity) $L_{LMC}$. $L_{LMC}$, similarly to our behavioural loss, is a probing based loss. Such losses have encountered success, for example in the context of HyperNetworks [7]. It however strongly differs from ours in its usage. In [1], authors use $L_{LMC}$ by computing the original model’s loss on point that lies linearly between the original neural network and the permuted neural network, so as to test for good linear mode connectivity. Ours differs in that, first, it does not need to compute the loss with regard to the original task, and therefore does not require any true labels. Second, its main interest lies in aligning the predictions made by the model and its reconstruction, and as such is closer to a soft-label distillation loss.
>
> In addition, the main contribution of our paper lies beyond simply adding a behavioural loss to the SANE [2] framework. Our submission presents a principled exploration of the duality of structure-based and behaviour-based signals in weight space representation learning, exploring valuable insights in why they synergise well and providing strong experimental validation results. We are convinced that the good results in [1] further strengthen and help generalise our main claim that there exists a strong synergy between structural and behavioural signals. Where [1] succeeds at leveraging dual structural and behavioural losses to solve the weights alignment problem, our paper attempts to show why those dual signals synergise well, giving insights into why their choice of losses has been particularly effective.
>
> We believe the discussion of the (di)similarities between [1] and our work to be a valuable addition to our manuscript. We have therefore incorporated a new related work paragraph that illustrates the differences between our approach and other probing-based ones that exist in the literature.
>
> > Limited experimental section. Given the fact that this work is mostly empirical, I expect the experimental section to be more extensive both in scale and learning setups. The authors focused on discriminative models and specifically only on small CNNs. There is no diversity in the reconstructed model architecture or the downstream tasks.
>
> We respectfully disagree with the reviewer that our work is mostly empirical. In Sections 2 and 3, we provide insights into why there is a need to capture features not adequately captured by structural reconstruction alone, and why the behavioural loss is a good candidate. Then, in our Section 4, we validate these hypotheses through empirical experiments, which show strong performance improvements across all downstream tasks and datasets. In addition, we also disagree that there is no diversity in the downstream tasks studied: we evaluated 2 different discriminative downstream tasks, a reconstructive downstream task, as well as a generative downstream task.
>
> We acknowledge, however, that a more extensive set of architectures could further strengthen our main claim. We address this point in more detail in the response to all reviewers, where we explain architectural challenges in scaling up the models we work on. In order to validate our findings on larger architectures, we have added a new Appendix D.5 where we run experiments on a new model zoo of LeNet-5 models trained on CIFAR-10, which are around 6 times larger that the models in our other zoos.
>
> > It would be interesting to see how significant are gamma and beta hyperparameters in the AE optimization.
>
> We have also addressed this point in the response to all reviewers. $\gamma$ has been set to the same value as in SANE [2]. We have added a new Appendix with the experiments validating our choice of hyperparameter $\beta$ using our validation sets.

---

> > ### Author Response · Authors · 2024-11-20
> > **Response to reviewer fxSa (2)**
> >
> > > The authors mentioned in the appendix that training with behavioral loss nearly doubles the training duration, which poses a challenge for large models.
> >
> > We have indeed addressed this in the Limitations section of our manuscript. On a positive note, Appendix D.4 indicates that even when using comparable computing resources, using both structure and behaviour still performs better than using only structure and running more epochs. We therefore do not consider the computational overhead to be a blocking issue.
> >
> > > Rather than using a common model zoo [3,4], the authors chose to create their own, which raises some questions.
> >
> > We would like to clarify that the zoos we use are extensions of the zoos in [3]. In [3], the zoos with variable hyperparameters use the activation function as a hyperparameter, which makes the computation of the behaviour loss tricky: it is actually more an architecture design choice than a hyperparameter of the optimisation algorithm. In this paper, we only consider fixed architectures. We therefore take a subset of the zoos in [3], fix some hyper-parameters and extend the values of hyper-parameters used for learning rate and weight decay. We keep hyperparameters the same across zoos for consistency purposes. Because of these slight alterations, we do not claim to use the zoos [3], but ours are directly derived from them.
> >
> > > The authors did not include confidence intervals in the experimental section. Moreover, in Tables 5 and 6 the authors provide additional results which are inclusive due to high std values. The same should be reported in the main text results.
> >
> > In Table 1, we compute the $R^2$ score over the whole test set, and therefore do not have any standard deviations to provide. In Table 2, we provide the maximum values across the set of models to illustrate the maximum accuracies reconstructed models can achieve, which is the main limitation when using structure only — the standard deviation of model accuracies would not be relevant in that context. Standard deviation is included for Tables 5 and 6, in the Appendix.
> >
> > Moreover, the claim that the results in Tables 5 and 6 are inconclusive due to high standard deviation values is not correct. One of our main claims is that adding the behavioural loss improves the agreement between models and their reconstructions. To support this, we compare the distribution of model agreements between the baseline ($\mathcal{L}_C + \mathcal{L}_S$) and the version that includes the behavioural loss ($\mathcal{L}_C + \mathcal{L}_S + \mathcal{L}_B$) in Table 5. A two-sided T-test for identical expectations shows statistical significance at the $10^{-9}$ level. The corresponding distributions can be visualised in Figure 2.
> >
> > > The only augmentation used for LC is a random permutation, why the authors did not use augmentations from [5,6]? [6] should be cited in this work.
> >
> > For this work, we limit ourselves to augmentations that do not alter the behaviour of the neural network, to ensure clean and meaningful signal to the behavioural loss. For this reason, we focus on random permutations. In future work, evaluating more augmentations specifically for behavioural losses, such as [6], may be an interesting field of study. To better reflect that, we have added the reference in our manuscript.
> >
> > > Based on the findings in Section 2, have the authors considered explicitly encouraging hybrid eigenvalue features? For example, they might incorporate a KL divergence loss relative to a predefined Beta distribution concentrated in the high and low values, effectively ignoring the intermediate range.
> >
> > That’s an interesting idea, which we haven’t considered. This ultimately merits exploration, but there are two caveats worth discussing.
> > 1. We understand the eigendecomposition as an indication for missing relevant features, not as the missing feature itself. Overfitting the metric may lead to unstable results.
> > 2. Penalising the distributions of eigenvalues before and after reconstruction requires computing the eigendecomposition at every iteration. That is quite costly, compared to inferencing the embedded model, particularly for large models.
> >
> > —
> >
> > [1-6] are identical to the reviewer’s reference numbers, for consistency purposes.
> >
> > [7] Ha et al., “HyperNetworks”, ICLR 2017

---

> > > ### Comment · Reviewer_fxSa · 2024-11-25
> > > **Reviewer response**
> > >
> > > I would like to thank the authors for their time and effort put in the rebuttal response. Reading the author's response, as well as other reviews, I have come to the conclusion that this paper does not meet the acceptance criteria, particularly due to its lack of novelty. My main concerens regarding this paper were not addressed during the rebuttal, therefore I will keep me socre.

---

### Official Review · Reviewer_6Tzb · 2024-11-03

**Soundness:** 3
**Presentation:** 3
**Contribution:** 3
**Rating:** 6
**Confidence:** 3

**Summary:**

This work proposes a new behavioral loss to enhance the performance of hypernetwork autoencoders trained to reconstruct neural network weights. It demonstrates improved results in reconstructing convolutional neural network weights on vision datasets. The study provides extensive validation of reconstruction quality, combining contrastive, structural, and behavioral losses to achieve this improvement.

**Strengths:**

* The paper is well-written and easy to follow, with clear, well-organized sections that ensure a smooth flow throughout. Additionally, the captions for tables and figures are self-contained, making it easier for readers to understand them in context.

* The results demonstrate a consistent improvement in both the performance of the reconstructed models and the richness of the information contained in the computed weight representations. The authors assess this information using linear probing to predict test accuracy and model generalization gaps, providing valuable insights into the learned representations.

* The authors have released the codebase for this work, which enhances the paper's reproducibility and allows others to verify and build upon their findings.

**Weaknesses:**

* The experimental setup is primarily focused on convolutional neural networks on three vision datasets without exploring other types of architectures or data. Including a broader range of architectures and datasets could further validate the approach’s generalizability. (See questions).

* Table 2 shows that the proposed behavioral loss improves performance across all three datasets. However, it would be insightful to examine whether the representations produced by the original CNNs are similar to those generated by the reconstructed CNNs. This analysis could further strengthen the findings, clarifying whether the generated weights yield representations closely aligned with the originals and thereby contribute to the observed performance improvements.

* The reported results do not include standard deviations, which could offer insight into the consistency and robustness of the proposed method across different runs. Including this information would enhance the reliability of the findings.

**Questions:**

* It would be valuable to explore how the proposed approach performs when applied to architectures other than CNNs, such as MLPs or larger models. This could provide further insights into the generalizability and robustness of the proposed method.

* It would be interesting to examine the effectiveness of the proposed method on other data modalities, such as text or graph data. Testing across diverse modalities could help determine the broader applicability and flexibility of the approach.

* It would be helpful to clarify whether all model zoos in the experiments were trained exclusively for image classification. Exploring the reconstruction of weights from models trained on other downstream tasks (e.g. image reconstruction) could provide insight into the versatility and task-specific performance of the method.

---

> ### Author Response · Authors · 2024-11-20
> **Response to reviewer 6Tzb**
>
> We would like to thank reviewer 6Tzb for their review, and will try to address the weaknesses and questions they raised with this response.
>
> > Table 2 shows that the proposed behavioral loss improves performance across all three datasets. However, it would be insightful to examine whether the representations produced by the original CNNs are similar to those generated by the reconstructed CNNs. This analysis could further strengthen the findings, clarifying whether the generated weights yield representations closely aligned with the originals and thereby contribute to the observed performance improvements.
>
> In the Appendix, we have included more results that space limitations prevented us to include in the maintext. In particular, Table 5 shows measures of structural and behavioural similarity between models and their reconstructions. These results report both averages and standard deviations, and show that there is a substantial increase in model agreement when using a behavioural loss. Does that correspond to what the reviewer had in mind?
>
> > The reported results do not include standard deviations, which could offer insight into the consistency and robustness of the proposed method across different runs. Including this information would enhance the reliability of the findings.
>
> We agree with the reviewer, and as such Table 5 includes both average and standard deviation for the structural and behavioural similarities of models and their reconstructions. For the discriminative downstream tasks, we compute the $R^2$ score over the whole test set, and therefore do not have any standard deviations to provide; results are however similar across all three datasets.
>
> > The experimental setup is primarily focused on convolutional neural networks on three vision datasets without exploring other types of architectures or data. Including a broader range of architectures and datasets could further validate the approach’s generalizability. (See questions).
>
> > It would be valuable to explore how the proposed approach performs when applied to architectures other than CNNs, such as MLPs or larger models. This could provide further insights into the generalizability and robustness of the proposed method.
>
> We acknowledge that including more architectures would help demonstrate the generalisability of our methods. We discuss in more detail in the response to all reviewers why scaling up to models in the million parameters range or larger is challenging. As discussed there, our work focuses more on validating the synergy between structural and behavioural signals, and we aim our experiments at validating these findings. We have added a new experiment to the Appendix that validates our findings on a LeNet-5 based architecture, strengthening the validity of our results on models ~6 times larger than those in our original experiments.
>
> > It would be interesting to examine the effectiveness of the proposed method on other data modalities, such as text or graph data. Testing across diverse modalities could help determine the broader applicability and flexibility of the approach.
>
> While we agree with the reviewer, we have chosen simpler models to facilitate experiment setup and comparison, and also because of limited compute resources.
>
> > It would be helpful to clarify whether all model zoos in the experiments were trained exclusively for image classification. Exploring the reconstruction of weights from models trained on other downstream tasks (e.g. image reconstruction) could provide insight into the versatility and task-specific performance of the method.
>
> All models have indeed been trained for image classification. However, since the proposed behavioural loss is using the $L^2$ distance between outputs rather than cross-entropy, it remains applicable to regression or reconstruction models as is.

---

> > ### Comment · Reviewer_6Tzb · 2024-11-26
> >
> > I want to thank the reviewer for their effort in clarifying my doubts and questions and I will keep my score.

---

### Author Response · Authors · 2024-11-20
**Response to all reviewers**

We would like to thank all reviewers for the work and thought they put in their constructive reviews. With this common response, we aim to address concerns and answer interrogations raised by multiple reviewers.

## Contribution meaningfulness

Several reviewers mention that our submission lacks technical novelty. We respectfully disagree: our work proposes a principled approach in studying the relationship between structural and behavioural signals in the context of weight-space representation learning, and therefore goes beyond simply adding a behavioural loss to the SANE framework [1]. Inspired by similar, recent works in the computer vision domain by Balestriero and LeCun [2], we first identify that in weight space, learning by reconstruction similarly omits features that are essential to reconstruct models with high test-set performance. We then derive a behavioural loss to directly target those missing features, and show how that loss differs from the structural one. Finally, we perform a set of experiments on a variety of downstream tasks in order to empirically validate these findings. As noted by several reviewers, evidence shown in our experiments is strong. We acknowledge that our submission proposes a quite simple solution, and that probing-based losses have been used in previous works, such as HyperNetworks [3]. We nevertheless believe our work to be a relevant addition to the weight space literature because it highlights and explores the importance of the dual structural and behavioural signal in a way that has not been done before, which in itself is of interest for the development of more advanced methods in the future.

## Architectures and task variety

In our submission, we choose to only test our results on three different model zoos composed of smaller CNNs. Reviewers have noted that adding more architectures, tasks or even modalities could make our results more convincing. We would first like to highlight that with this submission, we mostly aim to show the existence of a synergy between structural and behavioural signals in weight space. As such, our experiments are aimed at validating these findings, and they show strong empirical evidence towards this. Nevertheless, we do agree with reviewers that adding more architectures would strengthen our paper.

Unfortunately, as discussed in the Limitations, scaling up our method to larger models is highly non-trivial. Indeed, while SANE [1] can work on arbitrarily large models, it can do so by processing “windows” of weights. Computing the $L^2$ distance between a window and its reconstruction is straightforward, but computing their behaviour is not since it relies on the model as a whole, not only the window of weights considered. It is possible to find ways to improve the SANE framework to circumvent this issue, but this is out of scope for this paper where we only aim to validate our findings that there exists a synergy between structural and behavioural signals in weight space. For this reason, we limit our experiments to model sizes that can be processed as a whole by the Transformer-based SANE. Since the size complexity of Transformers increases with the square of the size of the context window, this limits us to relatively small models.

We have nevertheless run experiments on the CIFAR-10 dataset with larger CNNs based on the LeNet-5 architecture, with around 6 times more parameters than those presented in the original submission: results are very similar and also highlight a strong synergy between structure and behaviour. We have added these new experiments to the manuscript as Appendix D.5.

## Choice of hyperparameters $\gamma$ and $\beta$

Regarding $\gamma$, we made the choice to retain the hyperparameter’s value used by Schürholt et al. [1]. Since it is not the focus of this work, we have not explored multiple values of this hyperparameter. We have nevertheless studied the opportunity to set it to $0$ to deactivate the contrastive element of the loss in Appendix D.1.

Regarding $\beta$, we have tested multiple possible values on our validation sets, and kept the one that showed the most consistent performance. We recognise that this experiment is currently absent from the paper. We have added the corresponding experiment as Appendix D.6, which justifies the choice of the value $0.1$ for this hyperparameter.

—

[1] Schürholt et al., "Towards Scalable and Versatile Weight Space Learning.", ICML 2024.

[2] Balestriero and LeCun, “Learning by reconstruction produces uninformative features for perception”, ICML 2024.

[3] Ha et al., “HyperNetworks”, ICLR 2017

---

> ### Author Response · Authors · 2024-11-20
> **Revision changelog**
>
> We indicate here all major changes we made to the revised rebuttal submission, compared to the original submission.
>
> * Added an Appendix D.5 and Figure 10 in which we validate our methodology on a model zoo of larger LeNet-5 models trained on CIFAR-10
> * Added an Appendix D.6 as well as Tables 7 and 8, in which we discuss our choice of hyperparameter $\beta$ to $0.1$. Also added a reference to that Appendix in Section 4.1.
> * Added a new related work paragraph to incorporate other works using probing-based losses in the context of weight space learning
> * Added a new reference to “Improved Generalization of Weight Space Networks via Augmentations” by Shamsian et al. to the related work, and the related discussion.
> * Added a short definition of a “model zoo” at the beginning of Section 4.1

---

### Meta-Review · Area_Chair_dupz · 2024-12-21

**Metareview:**

The paper focuses on the weight space of neural networks as a new data modality, and posits that simply weight reconstruction is not enough for learning. Rather, a 'behavior loss' is needed to tie the learning to specific downstream tasks. The paper mainly focus on experiments with convolutional neural networks. Reviewers share the common opinion that the paper does not clarify well enough the difference with SANE. The authors clarify that the behavior loss is not their main contribution, however,  I agree that this can be better written, since it is even claimed as such in the abstract (' To address this issue, we propose a behavioral loss for training AEs in weight space.'). Given that clear novelty is a fundamental requirement to be published in a conference like ICLR, I suggest the authors to revise their manuscript and resubmit.

**Additional Comments On Reviewer Discussion:**

There were no significant comments or changes during the reviewer discussion.

---

### Decision · Program_Chairs · 2025-01-22

Reject